# ADVERSARIAL PERTURBATIONS CANNOT RELIABLY PROTECT ARTISTS FROM GENERATIVE AI

**Robert Hönig**
ETH Zurich

**Javier Rando**
ETH Zurich

**Nicholas Carlini**
Google DeepMind

**Florian Tramèr**
ETH Zurich

## ABSTRACT

Artists are increasingly concerned about advancements in image generation models that can closely replicate their unique artistic styles. In response, several protection tools against style mimicry have been developed that incorporate small adversarial perturbations into artworks published online. In this work, we evaluate the effectiveness of popular protections—with millions of downloads—and show they only provide a false sense of security. We find that low-effort and "off-the-shelf" techniques, such as image upscaling, are sufficient to create robust mimicry methods that significantly degrade existing protections. Through a user study, we demonstrate that *all existing protections can be easily bypassed*, leaving artists vulnerable to style mimicry. We caution that tools based on adversarial perturbations cannot reliably protect artists from the misuse of generative AI, and urge the development of alternative protective solutions.

## 1 INTRODUCTION

*Style mimicry* is a popular application of text-to-image generative models. Given a few images from an artist, a model can be finetuned to generate new images in that style (e.g., a spaceship in the style of Van Gogh). But style mimicry has the potential to cause significant harm if misused. In particular, many contemporary artists worry that others could now produce images that copy their unique art style, and potentially steal away customers (Heikkilä, 2022). As a response, several protections have been developed to protect artists from style mimicry (Shan et al., 2023a; Van Le et al., 2023; Liang et al., 2023). These protections add adversarial perturbations to images that artists publish online, in order to inhibit the finetuning process. These protections have received significant attention from the media—with features in the New York Times (Hill, 2023), CNN (Thorbecke, 2023) and Scientific American (Leffer, 2023)—and have been downloaded over 1M times (Shan et al., 2023a).

Yet, it is unclear to what extent these tools actually protect artists against style mimicry, especially if someone actively attempts to circumvent them (Radiya-Dixit et al., 2021). In this work, we show that state-of-the-art style protection tools—*Glaze* (Shan et al., 2023a), *Mist* (Liang et al., 2023) and *Anti-DreamBooth* (Van Le et al., 2023)—are ineffective when faced with simple *robust mimicry methods*. The robust mimicry methods we consider range from low-effort strategies—such as using a different finetuning script, or adding Gaussian noise to the images before training—to multi-step strategies that combine off-the-shelf tools. We validate our results with a user study, which reveals that robust mimicry methods can produce results indistinguishable in quality from those obtained from unprotected artworks (see Figure 1 for an illustrative example).

We show that existing protection tools merely provide a false sense of security. Our robust mimicry methods do not require the development of new tools or fine-tuneing methods, but only carefully combining standard image processing techniques *which already existed at the time that these protection tools were first introduced!*. Therefore, we believe that even low-skilled forgers could have easily circumvented these tools since their inception.

Although we evaluate specific protection tools that exist today, the limitations of style mimicry protections are inherent. Artists are necessarily at a disadvantage since they have to act first (i.e., once someone downloads protected art, the protection can no longer be changed). To be effective,

---

Code and images released at https://github.com/ethz-spylab/robust-style-mimicry.
Correspondence at {robert.hoenig, javier.rando, florian.tramer}@inf.ethz.ch

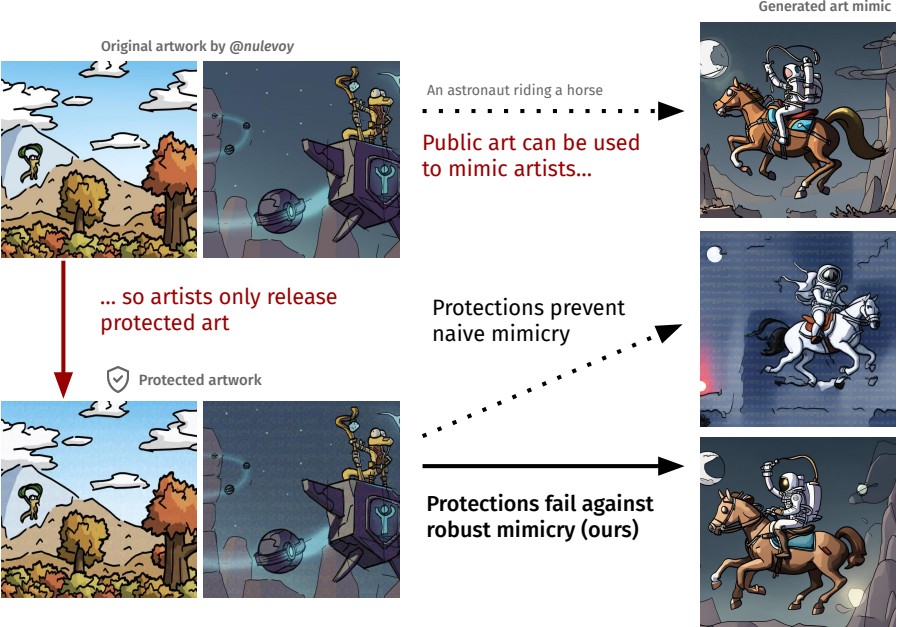

Figure 1: Artists are vulnerable to style mimicry from generative models finetuned on their art. Existing protection tools add small perturbations to published artwork to prevent mimicry (Shan et al., 2023a; Liang et al., 2023; Van Le et al., 2023). However, these protections fail against *robust mimicry methods*, giving a false sense of security and leaving artists vulnerable. Artwork by *@nulevoy* (Stas Voloshin), reproduced with permission.

protective tools face the challenging task of creating perturbations that transfer to *any* finetuning technique, even ones chosen adaptively in the future.[1] To illustrate this point, updated versions of Mist (Liang et al., 2023) and Glaze (Shan et al., 2023a) were released after the conclusion of our study, and yet we found these updated versions to be similarly ineffective against our methods. We thus caution that *adversarial machine learning techniques will not be able to reliably protect artists from generative style mimicry*, and urge the development of alternative measures to protect artists.

We disclosed our results to the affected protection tools prior to publication. In response, Glaze released a new version 2.1 that protects against the specific attacks we describe here.

## 2 BACKGROUND AND RELATED WORK

**Text-to-image diffusion models.** A latent diffusion model consists of an image autoencoder and a denoiser. The autoencoder is trained to encode and decode images using a lower-dimensional latent space. The denoiser predicts the noise added to latent representations of images in a diffusion process (Ho et al., 2020). Latent diffusion models can generate images from text prompts by conditioning the denoiser on image captions (Rombach et al., 2022). Popular text-to-image diffusion models include open models such as Stable Diffusion (Rombach et al., 2022) and Kandinsky (Razzhigaev et al., 2023), as well as closed models like Imagen (Saharia et al., 2022) and DALL-E (Ramesh et al.; Betker et al., 2023).

**Style mimicry.** Style mimicry uses generative models to create images matching a target artistic style. Existing techniques vary in complexity and quality (see Appendix G). An effective method is to finetune a diffusion model using a few images in the targeted style. Some artists worry that style mimicry can be misused to reproduce their work without permission and steal away customers (Heikkilä, 2022).

---

[1] A similar conclusion was drawn by Radiya-Dixit *et al.* (Radiya-Dixit et al., 2021), who argued that adversarial perturbations cannot protect users from facial recognition systems.

**Style mimicry protections.** Several tools have been proposed to prevent unauthorized style mimicry. These tools allow artists to include small perturbations—optimized to disrupt style mimicry techniques—in their images before publishing. The most popular protections are Glaze (Shan et al., 2023a) and Mist (Liang et al., 2023). Additionally, Anti-DreamBooth (Van Le et al., 2023) was introduced to prevent fake personalized images, but we also find it effective for style mimicry. Both Glaze and Mist target the encoder in latent diffusion models; they perturb images to obtain latent representations that decode to images in a different style (see Appendix H.1). On the other hand, Anti-DreamBooth targets the denoiser and maximizes the prediction error on the latent representations of the perturbed images (see Appendix H.2).

**Circumventing style mimicry protections.** Although not initially designed for this purpose, adversarial purification (Yoon et al., 2021; Shi et al., 2020; Samangouei et al., 2018) could be used to remove the perturbations introduced by style mimicry protections. DiffPure (Nie et al., 2022) is the strongest purification method and Mist claims robustness against it. Another existing method for purification is upscaling (Mustafa et al., 2019). Similarly, Mist and Glaze claim robustness against upscaling. Section 4.1 highlights flaws in previous evaluations and how a careful application of both methods can effectively remove mimicry protections.

IMPRESS (Cao et al., 2024) was the first purification method designed specifically to circumvent style mimicry protections. While IMPRESS claims to circumvent Glaze, the authors of Glaze critique the method's evaluation (Shan et al., 2023b), namely the reliance on automated metrics instead of a user study, as well as the method's poor performance on contemporary artists. Our work addresses these limitations by considering simpler and stronger purification methods, and evaluating them rigorously with a user study and across a variety of historical and contemporary artists. Our results show that the main idea of IMPRESS is sound, and that very similar robust mimicry methods are effective.

**Unlearnable examples .** Style mimicry protections build upon a line of work that aims to make data "unlearnable" by machine learning models (Shan et al., 2020; Huang et al., 2021; Cherepanova et al., 2021; Salman et al., 2023). These methods typically rely on some form of adversarial optimization, inspired by adversarial examples (Szegedy et al., 2013). Ultimately, these techniques always fall short of an *adaptive* adversary that enjoys a second-mover advantage: once unlearnable examples have been collected, their protection can no longer be changed, and the adversary can thereafter select a learning method tailored towards breaking the protections (Radiya-Dixit et al., 2021; Fowl et al., 2021; Tao et al., 2021).

## 3 THREAT MODEL

The goal of style mimicry is to produce images, of some chosen content, that mimic the style of a targeted artist. Since artistic style is challenging to formalize or quantify, we refrain from doing so and define a mimicry attempt as successful if it generates new images that a human observer would qualify as possessing the artist's style.

We assume two parties, the *artist* who places art online (e.g., in their portfolio), and a *forger* who performs style mimicry using these images. The challenge for the forger is that the artist first *protects* their original art collection before releasing it online, using a state-of-the-art protection tool such as Glaze, Mist or Anti-DreamBooth. We make the conservative assumption that *all* the artist's images available online are protected. If a mimicry method succeeds in this setting, we call it *robust*.

In this work, we consider style forgers who finetune a text-to-image model on an artist's images—the most successful style mimicry method to date (Shan et al., 2023a). Specifically, the forger finetunes a pretrained model $f$ on protected images $X$ from the artist to obtain a finetuned model $\hat{f}$. The forger has full control over the protected images and finetuning process, and can arbitrarily modify to maximize the mimicry success. Our *robust mimicry methods* combine a number of "off-the-shelf" manipulations that allow even low-skilled parties to bypass existing style mimicry protections. In fact, our most successful methods require only black-box access to a finetuning API for the model $f$, and could thus also be applied to proprietary text-to-image models that expose such an interface.

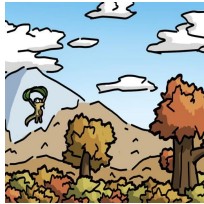 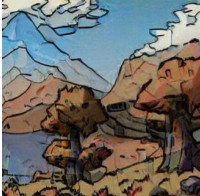 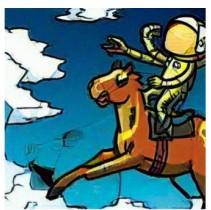 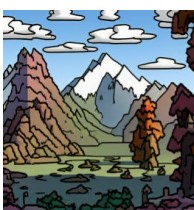 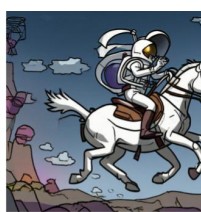

(a) Original artwork     (b) Finetuning used in (Shan et al., 2023a).          (c) Our finetuning

Figure 2: The protections of Glaze (Shan et al., 2023a) do not generalize across fine-tuning setups. We mimic the style of the contemporary artist @nulevoy from Glaze-protected images by using: (b) the finetuning script provided by Glaze authors; and (c) an alternative *off-the-shelf* finetuning script from HuggingFace. In both cases, we perform "naive" style mimicry with no effort to bypass Glaze's protections. Glaze protections are successful using finetuning from the original paper, but significantly degrade with our script. Our finetuning is also better for unprotected images (see Appendix D).

## 4 ROBUST STYLE MIMICRY

We say that a style mimicry method is *robust* if it can emulate an artist's style using only *protected* artwork. While methods for robust mimicry have already been proposed, we note a number of limitations in these methods and their evaluation in Section 4.1. We then propose our own methods (Section 4.3) and evaluation (Section 5) which address these limitations.

### 4.1 LIMITATIONS OF PRIOR ROBUST MIMICRY METHODS AND OF THEIR EVALUATIONS

**(1) Some mimicry protections do not generalize across finetuning setups.** Most forgers are inherently ill-intentioned since they ignore artists' genuine requests *not* to use their art for generative AI (Heikkilä, 2022). A successful protection must thus resist circumvention attempts from a reasonably resourced forger who may try out a variety of tools. Yet, in preliminary experiments, we found that Glaze (Shan et al., 2023a) performed significantly worse than claimed in the original evaluation, even before actively attempting to circumvent it. After discussion with the authors of Glaze, we found small differences between our off-the-shelf finetuning script, and the one used in Glaze's original evaluation (which the authors shared with us).[2] These minor differences in finetuning are sufficient to significantly degrade Glaze's protections (see Figure 2 for qualitative examples). Since our off-the-shelf finetuning script was not designed to bypass style mimicry protections, these results already hint at the superficial and brittle protections that existing tools provide: artists have no control over the finetuning script or hyperparameters a forger would use, so protections must be robust across these choices.

**(2) Existing robust mimicry attempts are sub-optimal.** Prior evaluations of protections fail to reflect the capabilities of moderately resourceful forgers, who employ state-of-the-art methods (even off-the-shelf ones). For instance, Mist (Liang et al., 2023) evaluates against *DiffPure* purifications using an outdated and low-resolution purification model. Using DiffPure with a more recent model, we observe significant improvements. Glaze (Shan et al., 2023a) is not evaluated against any version of DiffPure, but claims protection against *Compressed Upscaling*, which first compresses an image with JPEG and then upscales it with a dedicated model. Yet, we will show that by simply swapping the JPEG compression with Gaussian noising, we create *Noisy Upscaling* as a variant that is highly successful at removing mimicry protections (see Figure 26 for a comparison between both methods).

**(3) Existing evaluations are non-comprehensive.** Comparing the robustness of prior protections is challenging because the original evaluations use different sets of artists, prompts, and finetuning setups. Moreover, some evaluations rely on automated metrics (e.g., CLIP similarity) which are unreliable for measuring style mimicry (Shan et al., 2023a;b). Due to the brittleness of protection methods and the subjectivity of mimicry assessments, we believe a unified evaluation is needed.

---

[2]The two finetuning scripts mainly differ in the choice of library, model, and hyperparameters. We use a standard HuggingFace script and Stable Diffusion 2.1 (the model evaluated in the Glaze paper).

## 4.2 A Unified and Rigorous Evaluation of Robust Mimicry Methods

To address the limitations presented in Section 4.1, we introduce a unified evaluation protocol to reliably assess how existing protections perform against a variety of simple and natural robust mimicry methods. Our solutions to each of the numbered limitations above are: (1) The attacker uses a popular "off-the-shelf" finetuning script for the strongest open-source model that all protections claim to be effective for: Stable Diffusion 2.1. This finetuning script is chosen independently of any of these protections, and we treat it as a black-box. (2) We design four robust mimicry methods, described in Section 4.3. We prioritize simplicity and ease of use for low-expertise attackers by combining a variety of off-the-shelf tools. (3) We design and conduct a user study to evaluate each mimicry protection against each robust mimicry method on a common set of artists and prompts.

## 4.3 Our Robust Mimicry Methods

We now describe four robust mimicry methods that we designed to assess the robustness of protections. We primarily prioritize simple methods that only require *preprocessing* protected images. These methods present a higher risk because they are more accessible, do not require technical expertise, and can be used in black-box scenarios (e.g. if finetuning is provided as an API service). For completeness, we further propose one white-box method, inspired by IMPRESS (Cao et al., 2024).

We note that the methods we propose have been considered (at least in part) in prior work that found them to be *ineffective* against style mimicry protections (Shan et al., 2023a; Liang et al., 2023; Shan et al., 2023b). Yet, as we noted in Section 4.1, these evaluations suffered from a number of limitations. We thus re-evaluate these methods (or slight variants thereof) in a comprehensive manner and show that they are significantly more successful than previously claimed.

**Black-box preprocessing methods.**

✦ *Gaussian noising.* As a simple preprocessing step, we add small amounts of Gaussian noise to protected images. This approach can be used ahead of any black-box diffusion model.

✦ *DiffPure.* We use image-to-image models to remove perturbations introduced by the protections, also called DiffPure (Nie et al., 2022) (see Appendix I.1). This method is black-box, but requires two different models: the purifier, and the one used for style mimicry. We use Stable Diffusion XL as our purifier.

✦ *Noisy Upscaling.* We introduce a simple and effective variant of the two-stage upscaling purification considered in Glaze (Shan et al., 2023a). Their method first performs JPEG compression (to minimize perturbations) and then uses the Stable Diffusion Upscaler (Rombach et al., 2022) (to mitigate degradations in quality). Yet, we find that upscaling actually *magnifies* JPEG compression artifacts instead of removing them. To design a better purification method, we observe that the Upscaler is trained on images augmented with Gaussian noise. Therefore, we purify a protected image by first applying Gaussian noise and then applying the Upscaler. This Noisy Upscaling method introduces no perceptible artifacts and significantly reduces protections (see Figure 26 for an example and Appendix I.2 for details).

**White-box methods.**

✦ *IMPRESS++.* For completeness, we design a white-box method to assess whether more complex methods can further enhance the robustness of style mimicry. Our method builds on IMPRESS (Cao et al., 2024) but adopts a different loss function and further applies *negative prompting* (Miyake et al., 2023) and *denoising* to improve the robustness of the sampling procedure (see Appendix I.3 and Figure 27 for details).

## 5 Experimental Setup

**Protection tools.**   We evaluate three protection tools—Mist, Glaze and Anti-DreamBooth—against four robust mimicry methods—Gaussian noising, DiffPure, Noisy Upscaling and IMPRESS++—and a baseline mimicry method. We refer to a combination of a protection tool and a mimicry method as

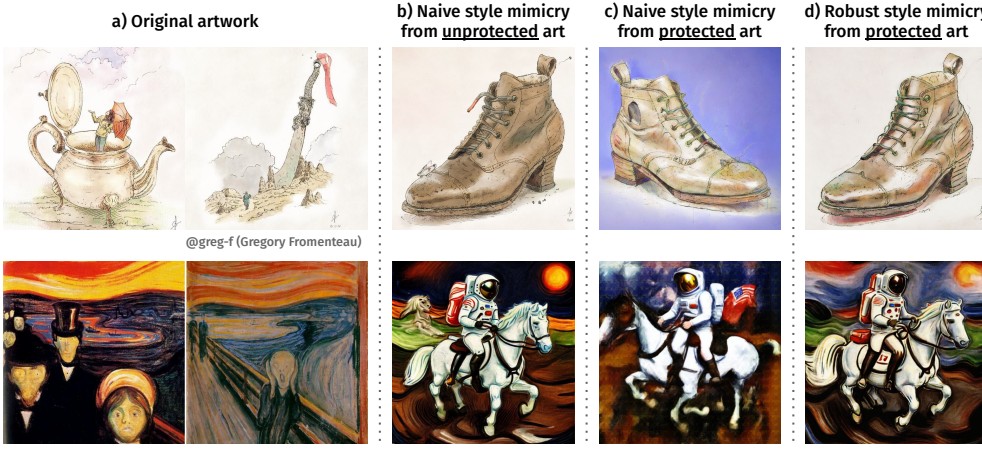

Figure 3: Examples of robust style mimicry for two different artists: @greg-f (contemporary) and Edvard Munch (historical). Cherry-picked examples with strong protections and successful robust mimicry. We apply Noisy Upscaling for prompts: "*a shoe*" and "*an astronaut riding a horse*".

a *scenario*. We thus analyze fifteen possible scenarios. Appendix J describes our experimental setup for style mimicry and protections in detail.

**Artists.** We evaluate each style mimicry scenario on images from 10 different artists, which we selected to maximize style diversity. To address limitations in prior evaluations (Shan et al., 2023b), we use five historical artists as well as five contemporary artists who are unlikely to be highly represented in the generative model's training set (two of these were also used in Glaze's evaluation).[3] All details about artist selection are included in Appendix J.

**Implementation.** Our mimicry methods finetune Stable Diffusion 2.1 (Rombach et al., 2022), the best open-source model available at the time when the protections we study were introduced. We use an off-the-shelf finetuning script from HuggingFace (see Appendix J.1 for details). We first validate that our style mimicry pipeline is successful on unprotected art using a user study, detailed in Appendix K.1. For protections, we use the original codebases to reproduce Mist and Anti-Dreambooth. Since Glaze does not have a public codebase (and the authors were unable to share one), we use the released Windows application binary (version 1.1.1) as a black-box. We set each scheme's hyperparameters to maximize protections. See Appendix J.2 for details on the configuration for each protection.

We perform robust mimicry by finetuning on 18 different images per artist. We then generate images for 10 different prompts. These prompts are designed to cover diverse motifs that the base model, Stable Diffusion 2.1, can successfully generate. See Appendix K for details about prompt design.

**User study.** To measure the success of each style mimicry scenario, we rely only on human evaluations since previous work found automated metrics (e.g., using CLIP (Radford et al., 2021)) to be unreliable (Shan et al., 2023a;b). Moreover, style protections not only prevent style transfer, but also reduce the overall quality of the generated images (see Figure 3 for examples). We thus design a user study to evaluate image quality and style transfer as independent attributes of the generations.[4]

We acknowledge that an ideal study would recruit artists, as was done in (Shan et al., 2023a). Unfortunately, most artists we reached out to were reluctant to participate in a study that shows limitations of existing protective tools (a small number of artists did acknowledge the success of our

---

[3]Contemporary Artists were selected from *Artstation*. We keep them anonymous throughout this work—and refrain from showcasing their art—except for artists who gave us explicit permission to share their identity and art. We will share all images used in our experiments upon request with researchers.

[4]The user study was approved by our institution's IRB.

methods when targeting their art styles, but they did not form a large enough cohort to get statistically significant results).

Our user study therefore relies on Amazon Mechanical Turk (MTurk) annotators, with stringent measures taken to ensure the quality and reliability of responses (see Appendix K). Our study asks participants to compare image pairs, where one image is generated by a robust mimicry method, and the other from a baseline state-of-the-art mimicry method that uses *unprotected* art of the artist. A perfectly robust mimicry method would generate images of quality and style indistinguishable from those generated directly from unprotected art. We perform two separate studies: one assessing image quality (e.g., which image looks "better") and another evaluating stylistic transfer (i.e., which image captures the artist's original style better, disregarding potential quality artifacts). Our results show that these two metrics obtain very similar results across all scenarios. Appendix K describes our user study and interface in detail.

As noted by the authors of Glaze (Shan et al., 2023a), the users of platforms like MTurk might not have high artistic expertise. However, we believe that the judgment of non-artists is also relevant as they may ultimately represent potential *consumers* of digital art. Thus, if lay people consider mimicry attempts to be successful, mimicked art could hurt an artist's business. Also, to mitigate potential issues with the quality of annotations (Kennedy et al., 2020), we put in place several control mechanisms to filter out low-quality annotations to the best of our abilities (details in Appendix K). Furthermore, as noted above, a small number of artists did acknowledge that they found our methods effective.

**Evaluation metric.** We define the *success rate* of a robust mimicry method as the percentage of annotators (5 per comparison) who prefer outputs from the robust mimicry method over those from a baseline method finetuned on *unprotected* art (when judging either style match or overall image quality). Formally, we define the success rate for an artist in a specific scenario as:

$$\texttt{success rate} = \frac{1}{10 \cdot 5} \sum_{\text{prompt}}^{10} \sum_{\text{annotator}}^{5} \mathbb{1}\big[\textit{robust mimicry} \text{ preferred over } \textit{unprotected mimicry}\big]$$

(1)

A perfectly robust mimicry method would thus obtain a success rate of 50%, indicating that its outputs are indistinguishable in quality and style from those from the baseline, unprotected method. In contrast, a very successful protection would result in success rates of around 0% for robust mimicry methods, indicating that mimicry on top of protected images always yields worse outputs.

## 6 RESULTS

In Figure 4, we report the distribution of success rates per artist (N=10) for each scenario. We averaged the quality and stylistic transfer success rates to simplify the analysis (detailed results can be found in Appendix C). Since the forger can try multiple mimicry methods for each prompt, and then decide which one worked best, we also evaluate a "best-of-4" method that picks the most successful mimicry method for each generation (according to human evaluators). Best-of-4 also also illustrates how different methods succeed for different styles and artists, as it outperforms all independent methods.

### 6.1 MAIN FINDINGS: ALL PROTECTIONS ARE EASILY CIRCUMVENTED

We find that all existing protective tools create a false sense of security and leave artists vulnerable to style mimicry. Indeed, our best robust mimicry methods produce images that are, on average, indistinguishable from baseline mimicry attempts using unprotected art. Since many of our simple mimicry methods only use tools that were available before the protections were released, style forgers may have already circumvented these protections since their inception.

Noisy upscaling is the most effective method for robust mimicry, with a median success rate above 40% for each protection tool (recall that 50% success indicates that the robust method is indistinguishable from a mimicry using unprotected images). This method only requires preprocessing images and black-box access to the model via a finetuning API. Other simple preprocessing methods

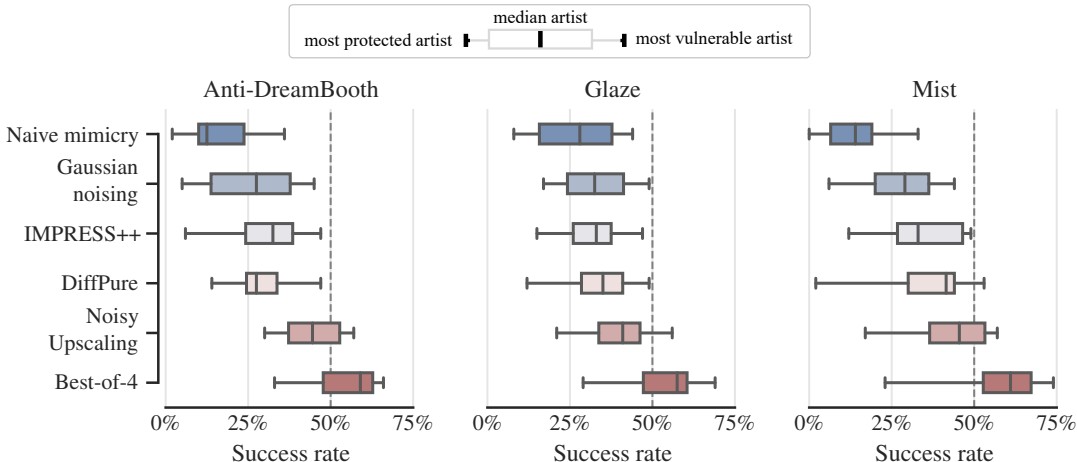

Figure 4: Success rate per artist (N=10) on all mimicry scenarios. Box plots represent success rates for most protected, quartiles, median and least protected artists, respectively. Success rates around 50% indicate that robust mimicry outputs are indistinguishable in style and quality from mimicry outputs based on unprotected images. *Best-of-4* selects the most successful method for each prompt.

like Gaussian noising or DiffPure also significantly reduce the effectiveness of protections. The more complex white-box method IMPRESS++ does not provide significant advantages. Sample generations for each method are in Appendix B.

A style forger does not have to use a single robust mimicry method, but can test all of them and select the most successful. This "best-of-4" approach always beats the baseline mimicry method over unprotected images (which attempts a single method and not four) for all protections.

Appendix A shows images at each step of the robust mimicry process (i.e., protections, preprocessing, and sampling). Appendix B shows example generations for each protection and mimicry method. Appendix C has detailed success rates broken down per artist, for both image style and quality.

## 6.2 ANALYSIS

We now discuss key insights and lessons learned from these results.

**Glaze protections break down without any circumvention attempt.** Results for Glaze without robust mimicry (see "Naive mimicry" row in Figure 4) show that the tool's protections are often ineffective. Without any robustness intervention, 30% of the images generated with our off-the-shelf finetuning are rated as better than the baseline results using only unprotected images. This contrasts with Glaze's original evaluation, which claimed a success rate of at most 10% for robust mimicry.[5] This difference is likely due to the protection's brittleness to slight changes in the finetuning setup (as we illustrated in Section 4.1). With our best robust mimicry method (noisy upscaling) the median success rate across artists rises further to 40%, and our best-of-4 strategy yields results indistinguishable from the baseline for a majority of artists.

**Robust mimicry works for contemporary and historical artists alike.** Shan et al. (2023b) note that one of IMPRESS' main limitations is that "purification has a limited effect when tested on artists that are not well-known historical artists already embedded in original training data". Yet, we find that our best-performing robust mimicry method—Noisy Upscaling—has a similar success rate for historical artists (42.2%) and contemporary artists with little representation in the model's training set (43.5%).

---

[5]The original evaluation in Glaze directly asks annotators whether a mimicry is successful or not, rather than a binary comparison between a robust mimicry and a baseline mimicry as in our setup. Shan et al. (2023a) report that mimicry fails in 4% of cases for unprotected images, and succeeds in 6% of cases for protected images. This bounds the success rate for robust mimicry—according to our definition in Equation (1)—by at most 10%.

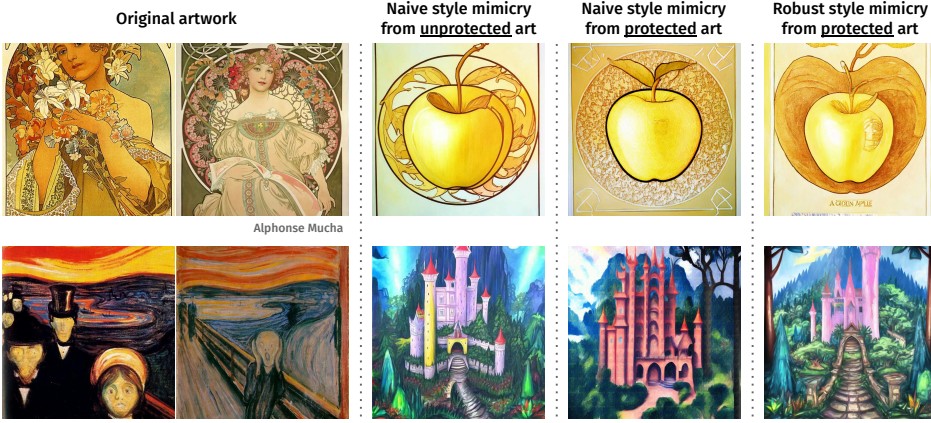

Figure 5: Randomly selected comparisons where all 5 annotators preferred mimicry from unprotected art over robust mimicry. Both use Noisy Upscaling for robust mimicry.

**Protections are highly non-uniform across artists.** As we observe from Figure 4, the effectiveness of protections varies significantly across artists: the least vulnerable artist (left-most whisker) enjoys much stronger mimicry protections than the median artist or the most vulnerable artist (right-most whisker). We find that robust mimicry is the least successful for artists where the baseline mimicry from unprotected images gives poor results to begin with (cf. results for artist $A_1$ in Appendix C and Appendix K.1). Yet, since existing tools do not provide artists with a way to *check* how vulnerable they are, these tools still provide a false sense of security for all artists. This highlights an inherent asymmetry between protection tools and mimicry methods: protections should hold for *all* artists alike, while a mimicry method might successfully target only specific artists.

**Robust mimicry failures still remove protection artifacts.** We manually checked the cases where all annotators ranked mimicry from unprotected art as better than robust mimicry with Noisy Upscaling. Figure 5 shows two examples. We find that in many instances, the model fails to mimic the style accurately even from unprotected art. In these cases, robust mimicry is still able to generate clear images that are similar to unprotected mimicry, but neither matches the original style well.

## 7    DISCUSSION AND BROADER IMPACT

**Adversarial perturbations do not protect artists from style mimicry.** Our work is not intended as an exhaustive search for the best robust mimicry method, but as a demonstration of the brittleness of existing protections. Because these protections have received significant attention, artists may believe they are effective. But our experiments show *they are not*. As we have learned from adversarial ML, whoever acts first (in this case, the artist) is at a fundamental disadvantage (Radiya-Dixit et al., 2021). We urge the community to acknowledge these limitations and think critically when performing future evaluations.

**Just like adversarial examples defenses, mimicry protections should be evaluated adaptively.** In adversarial settings, where one group wants to prevent another group from achieving some goal, it is necessary to consider "adaptive attacks" that are specifically designed to evade the defense (Carlini & Wagner, 2017). Unfortunately, as repeatedly seen in the literature on machine learning robustness, even after adaptive attacks were introduced, many evaluations remained flawed and defenses were broken by (stronger) adaptive attacks (Tramer et al., 2020). We show it is the same with mimicry protections: simple adaptive attacks significantly reduce their effectiveness. Surprisingly, most protections we study claim robustness against input transformations (Liang et al., 2023; Shan et al., 2023a), but minor modifications were sufficient to circumvent them.

We hope that the literature on style mimicry prevention will learn from the failings of the adversarial example literature: performing reliable, future-proof evaluations is much harder than proposing a

new defense. Especially when techniques are widely publicized in the popular press, we believe it is necessary to provide users with exceptionally high degrees of confidence in their efficacy.

**Protections are broken from day one, and cannot improve over time.** Our most successful robust style mimicry methods rely solely on techniques that existed before the protections were introduced. Also, protections applied to online images cannot easily be changed (i.e., even if the image is perturbed again and re-uploaded, the older version may still be available in an internet archive) (Radiya-Dixit et al., 2021). It is thus challenging for a broken protection method to be fixed retroactively. Of course, an artist can apply the new tool to their images going forward, but pre-existing images with weaker protections (or none at all) will significantly boost an attacker's success (Shan et al., 2023a).

Nevertheless, the Glaze and Mist protection tools recently received significant updates (after we had concluded our user study). Yet, we find that the newest 2.0 versions do not protect against our robust mimicry attempts either (see Appendix E and F). A subsequent version of Glaze (2.1) explicitly targets the methods we studied, but this does not change the fact that all previously protected art remains vulnerable, and that future attacks could again attempt to adaptively evade the newest protections. The same holds true for attempts to design similar protections for other data modalities, such as video (Passananti et al., 2024) or audio (Gokul & Dubnov, 2024).

**Ethics and broader impact.** The goal of our research is to help artists better decide how to protect their artwork and business. We do not focus on creating the *best* mimicry method, but rather on highlighting limitations in popular perturbation tools—especially since using these tools incurs a cost, as they degrade the quality of published art. We disclose our results to the affected protection tools prior to publication, so that they can determine the best course of action for their users.

Further, insecure protection tools may mislead artists to believe it is safe to release their work, enabling forgery and putting them in a worse situation than if they had been more cautious in the absence of any protection. With this work, we hope to raise awareness among artists about the fundamental limitations of protection tools.

With respect to our paper, all the art featured in this paper comes either from historical artists, or from contemporary artists who explicitly permitted us to display their work. We hope our results will inform improved non-technical protections for artists in the era of generative AI.

**Limitations and future work.** A larger study with more than 10 artists and more annotators may help us better understand the difference in vulnerability across artists. The protections we study are not designed in awareness of our robust mimicry methods. However, we do not believe this limits the extent to which our general claims hold: artists will always be at a disadvantage if attackers can design adaptive methods to circumvent the protections.

## ACKNOWLEDGEMENTS

We thank all the MTurkers that engaged with our tasks, especially those that provided valuable feedback during our preliminary studies to improve the survey. We thank the contemporary artists Stas Voloshin (@nulevoy) and Gregory Fromenteau (@greg-f) for allowing us to display their artwork in this paper. JR is supported by an ETH AI Center doctoral fellowship.

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

## A DETAILED ART EXAMPLES

This section illustrates how images look like at every stage of our work. We include (1) original artwork from a contemporary artist (@nulevoy)[6] as a reference in Figure 6, (2) the original artwork after applying each of the available protections in Figure 7, (3) one image after applying the cross product of all protections and preprocessing methods in Figure 8, (4) baseline generations from a model trained on unprotected art in Figure 9, and (5) robust mimicry generations for each scenario in Figure 10.

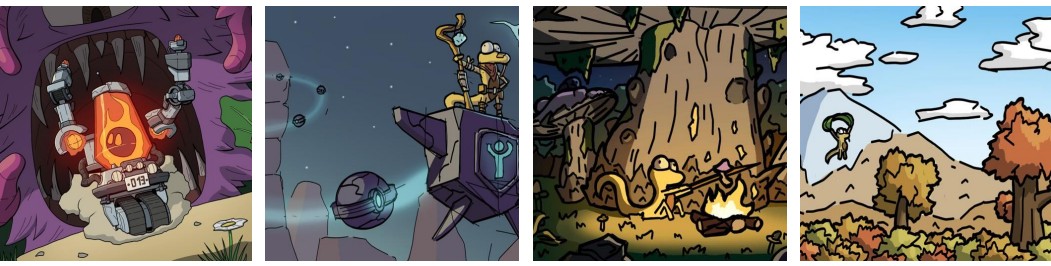

Figure 6: 4 samples from the original artwork from @nulevoy.

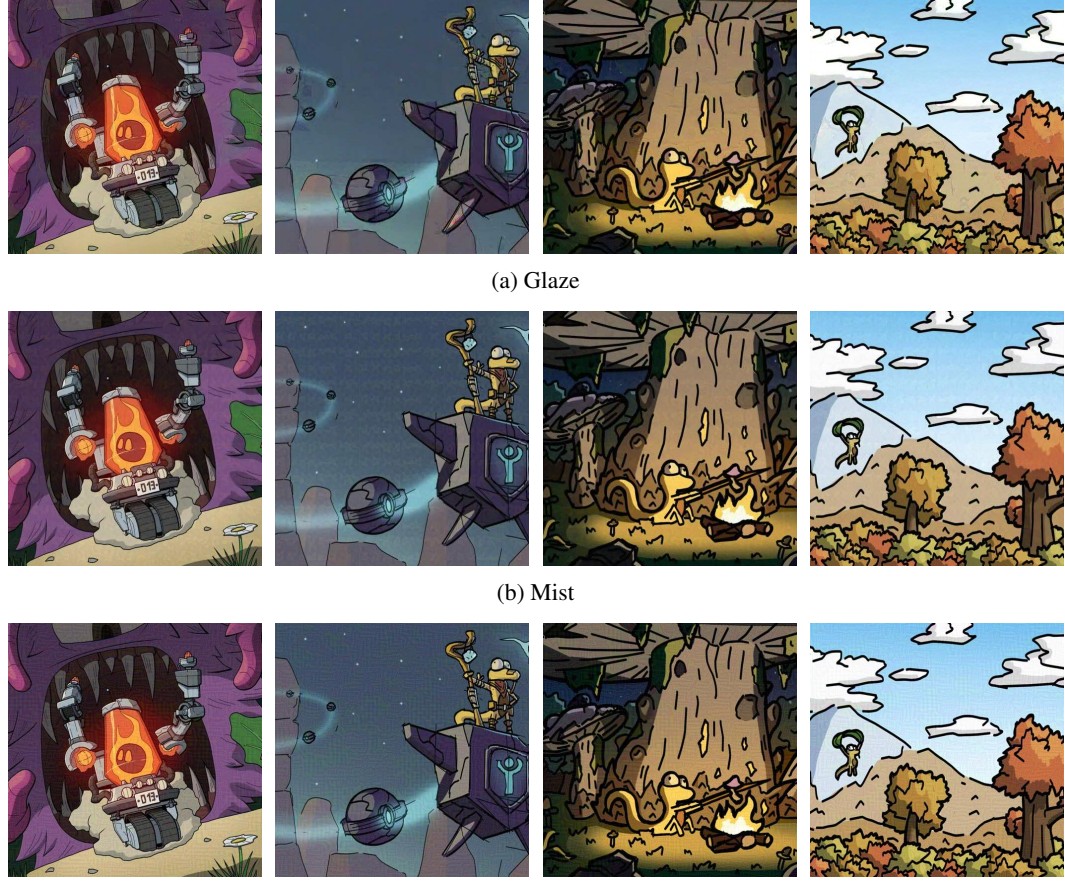

(a) Glaze

(b) Mist

(c) Anti-DreamBooth

Figure 7: Artwork in Figure 6 after applying different protections.

---

[6]The artist gave explicit permission for the use of their art

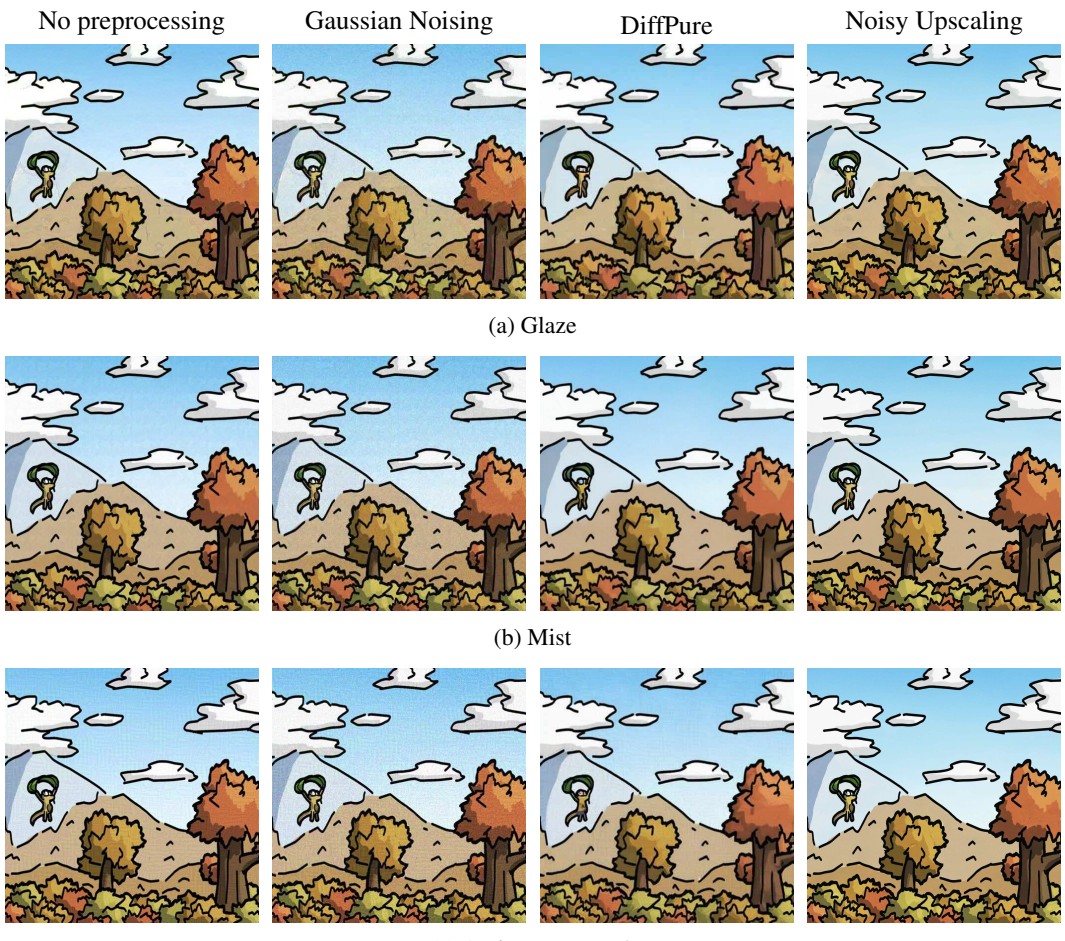

Figure 8: Artwork used for finetuning after applying preprocessing methods to protected images in Figure 7. Each row represents a protection, and each column a preprocessing method. Noisy Upscaling is the most successful preprocessing technique at removing the perturbations introduced by protections.

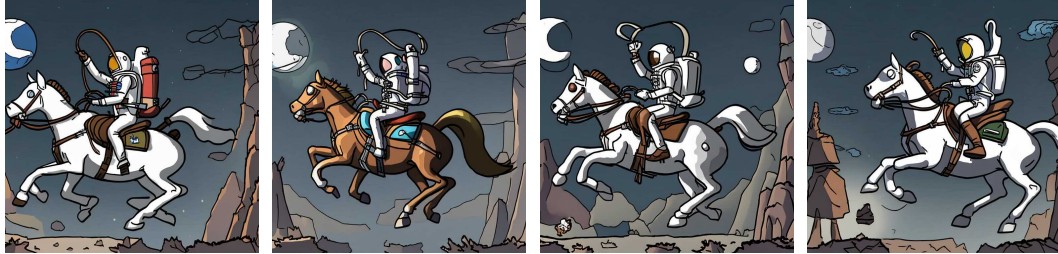

Figure 9: Generations in the style of @nulevoy after finetuning on *unprotected* images. Each generation is sampled with a different seed.

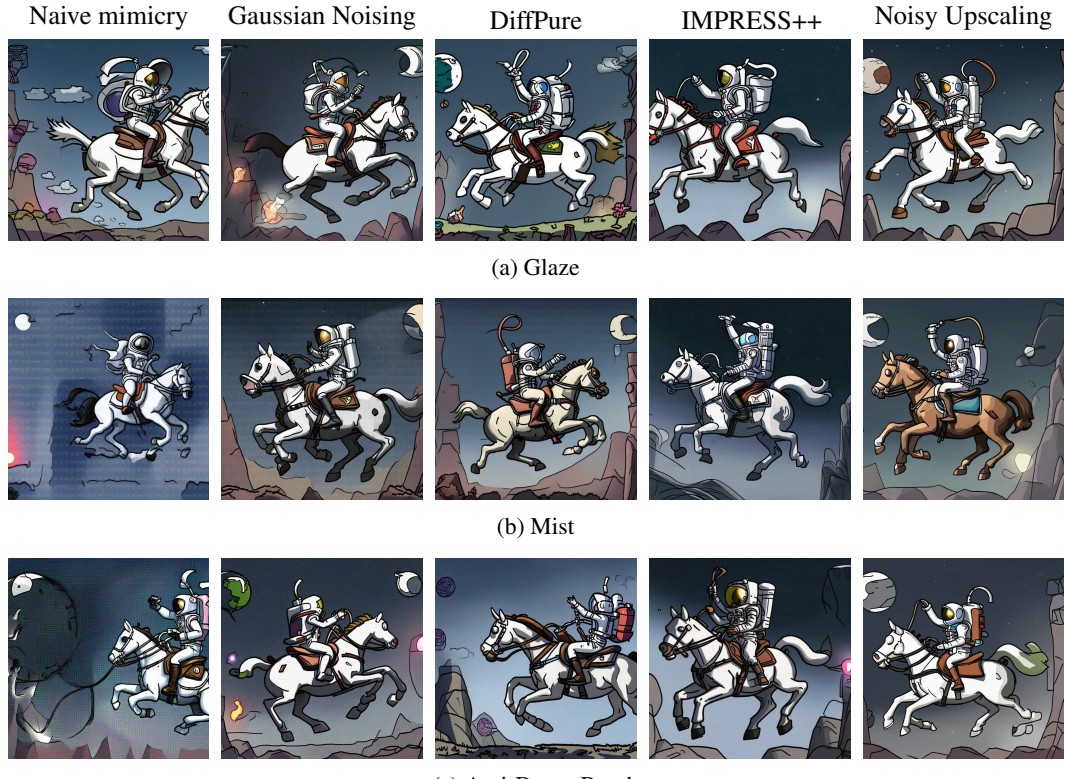

Figure 10: Generations in the style of @nulevoy using robust mimicry methods for the prompt "*an astronaut riding a horse*". Each row represents which protection was applied to the finetuning data. Each column represents the robust mimicry method used. The first column indicates naive mimicry was applied (i.e. we trained directly on the protected images). Figure 9 includes sample generations from a model trained on artwork without protections.

# B  ROBUST MIMICRY GENERATIONS

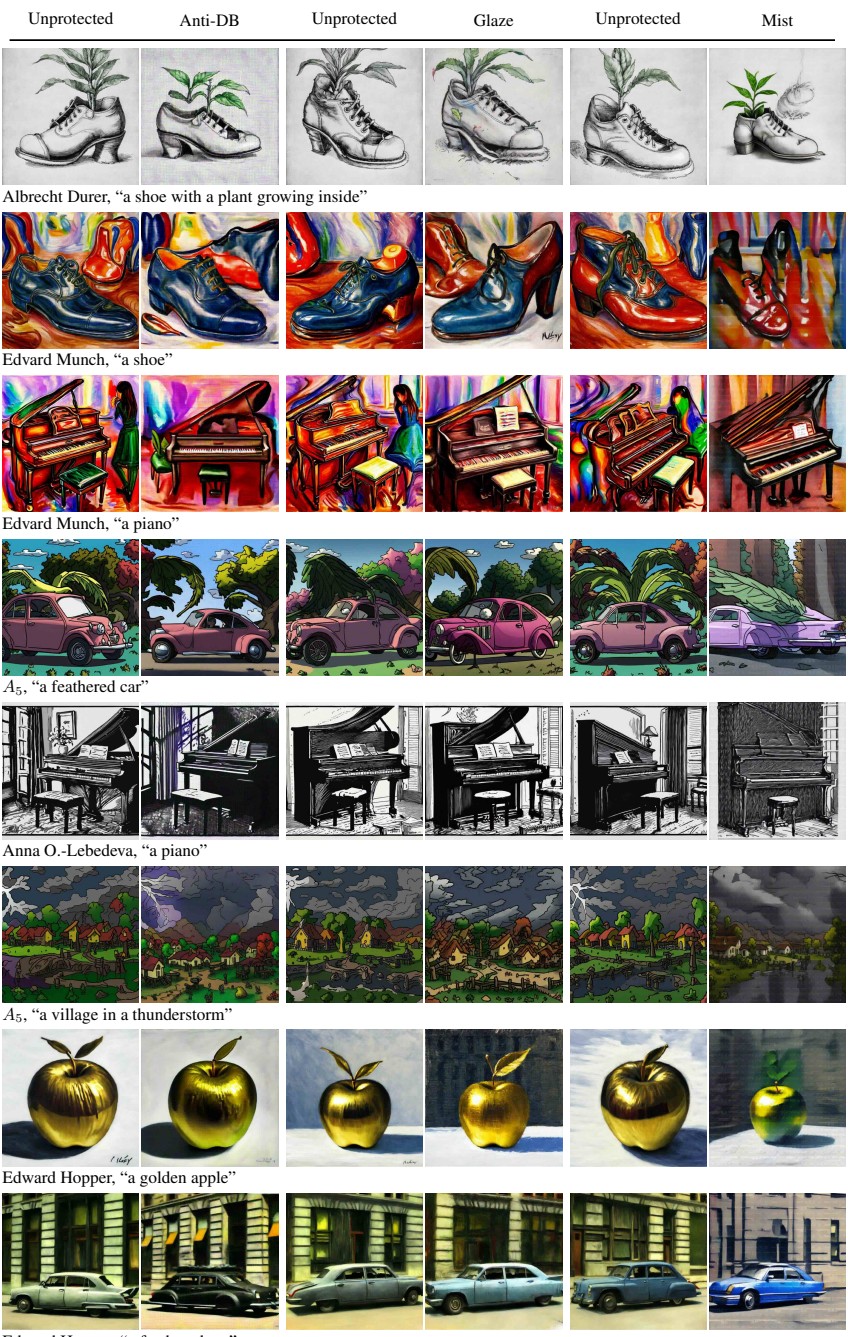

Figure 11: Style mimicry for all protections using *naive mimicry*—no robust method is used and we finetune directly on protected images. We randomly chose artists and prompts. Each image pair shows the protected generation and generation from unprotected art.

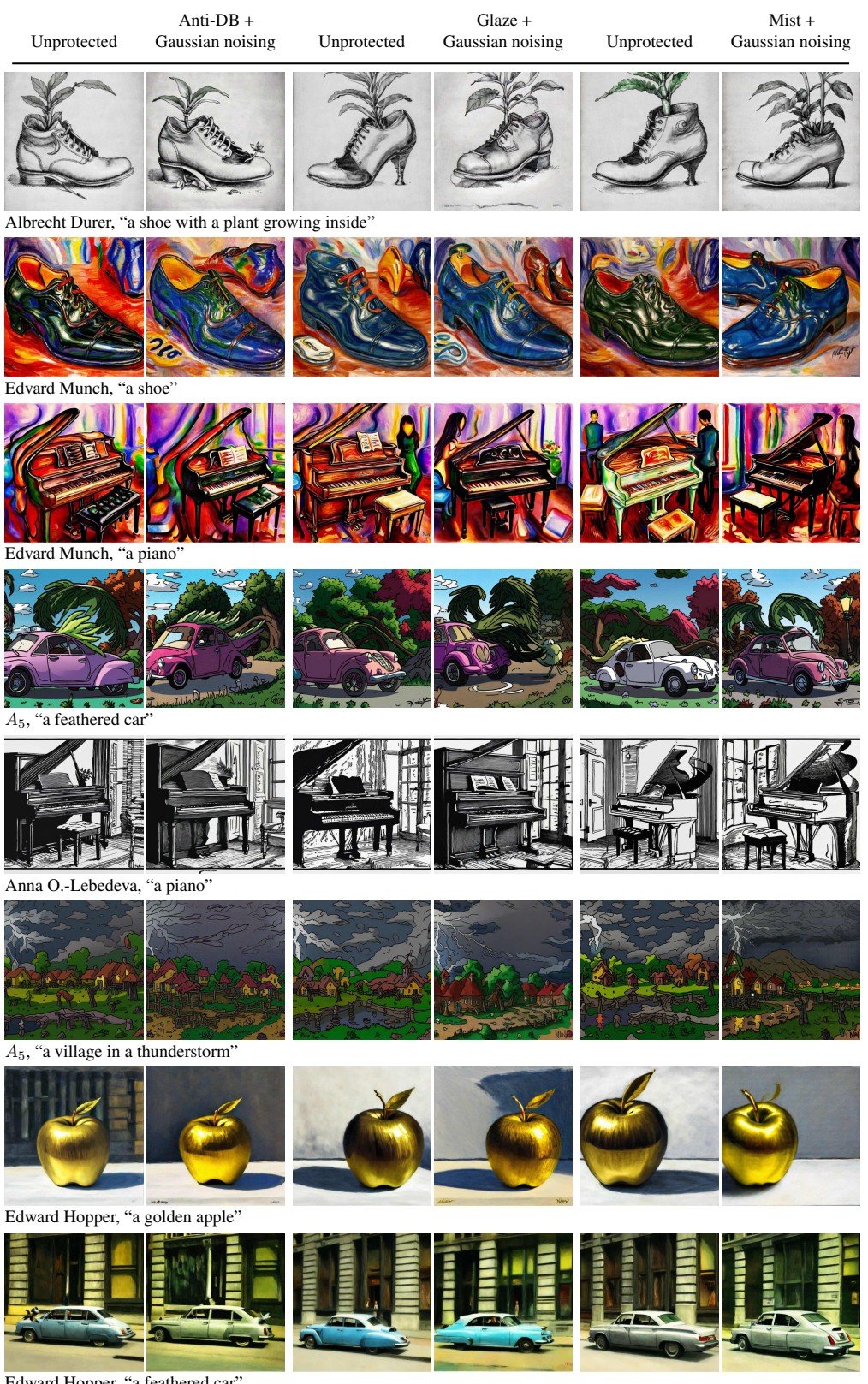

Figure 12: Style mimicry for all protections using *Gaussian Noising*. We randomly chose artists and prompts. Each image pair shows the protected robust generation and generation from unprotected art.

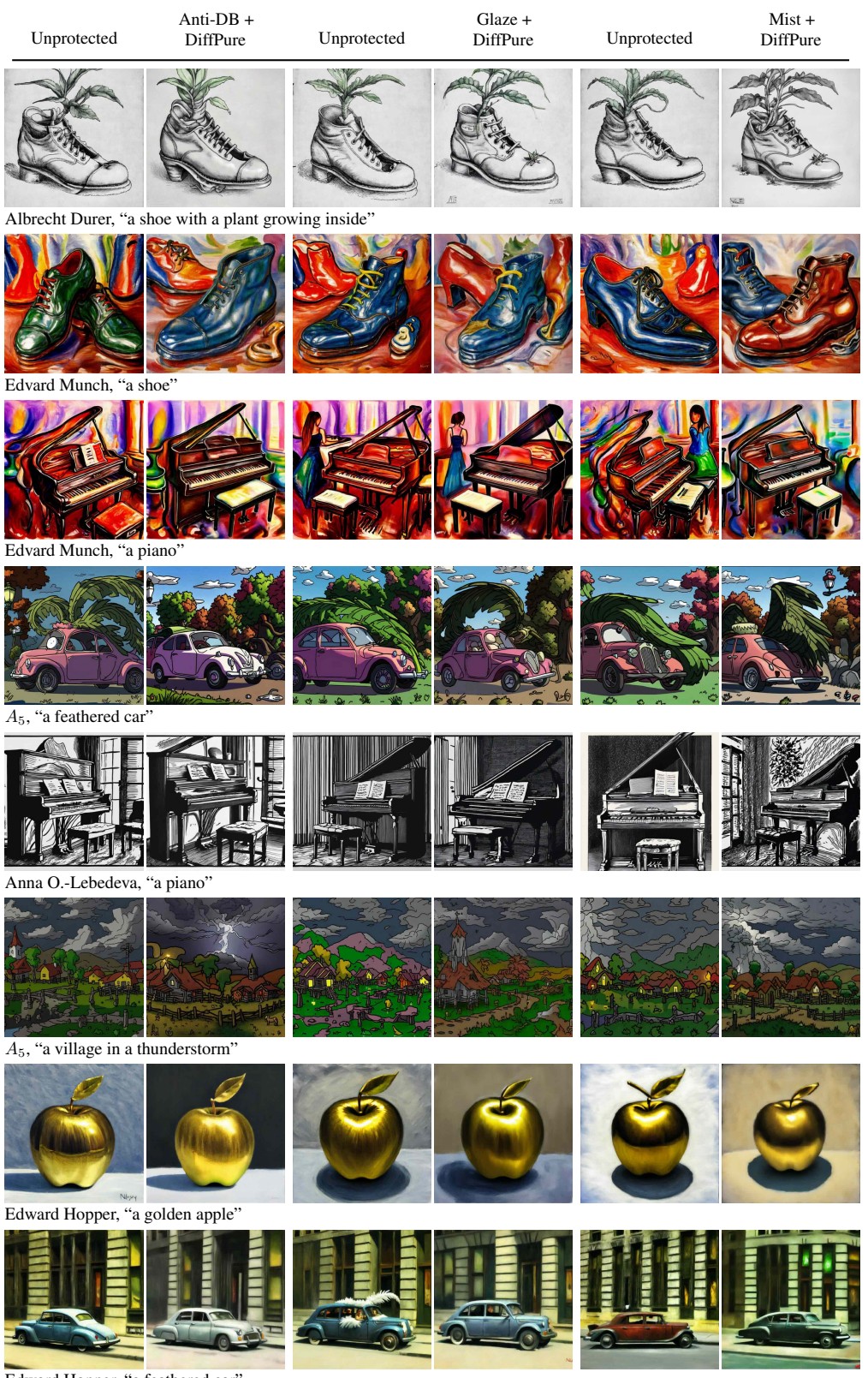

Figure 13: Style mimicry for all protections using *DiffPure*. We randomly chose artists and prompts. Each image pair shows the protected robust generation and generation from unprotected art.

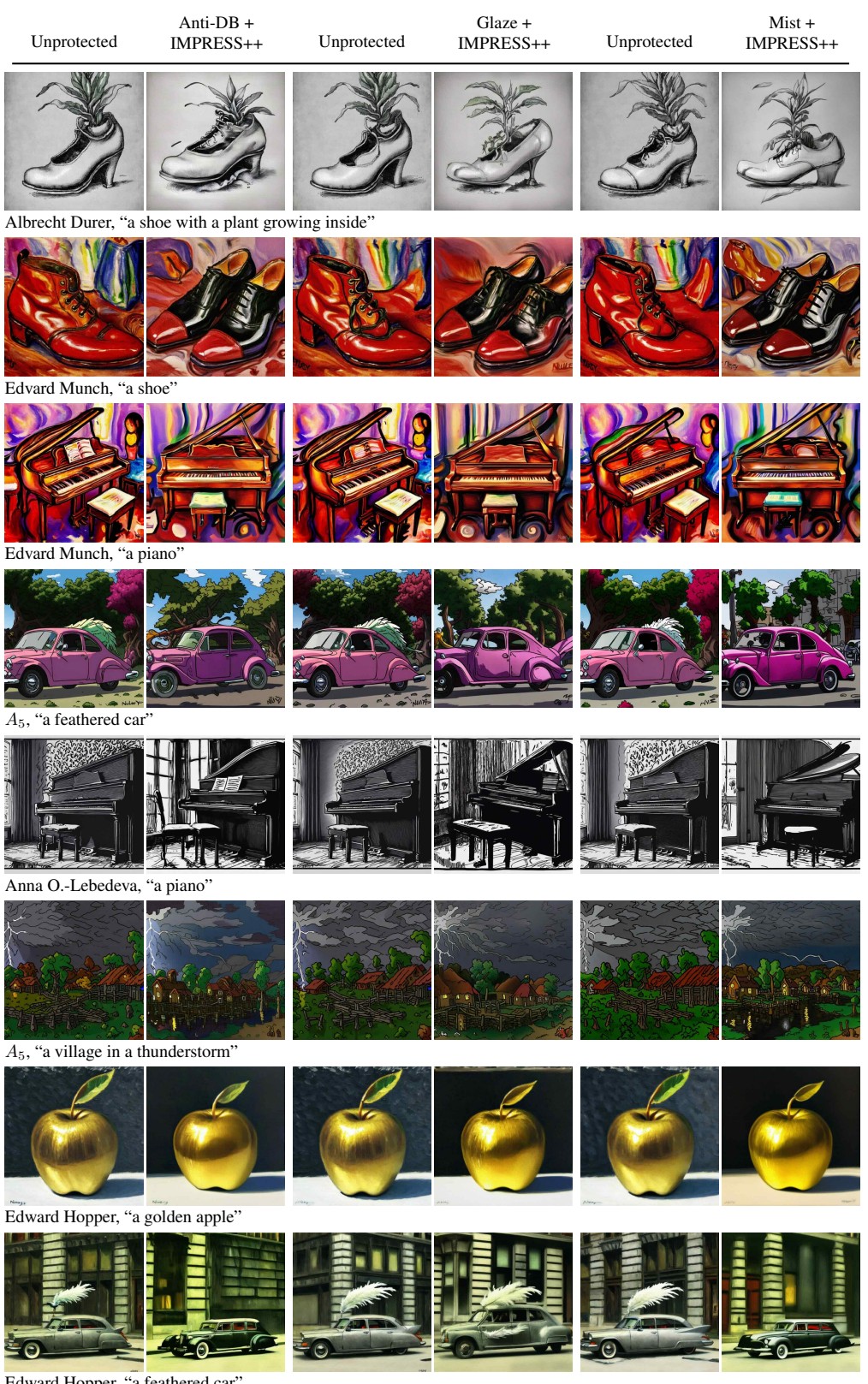

Figure 14: Style mimicry for all protections using *IMPRESS++*. We randomly chose artists and prompts. Each image pair shows the protected robust generation and generation from unprotected art.

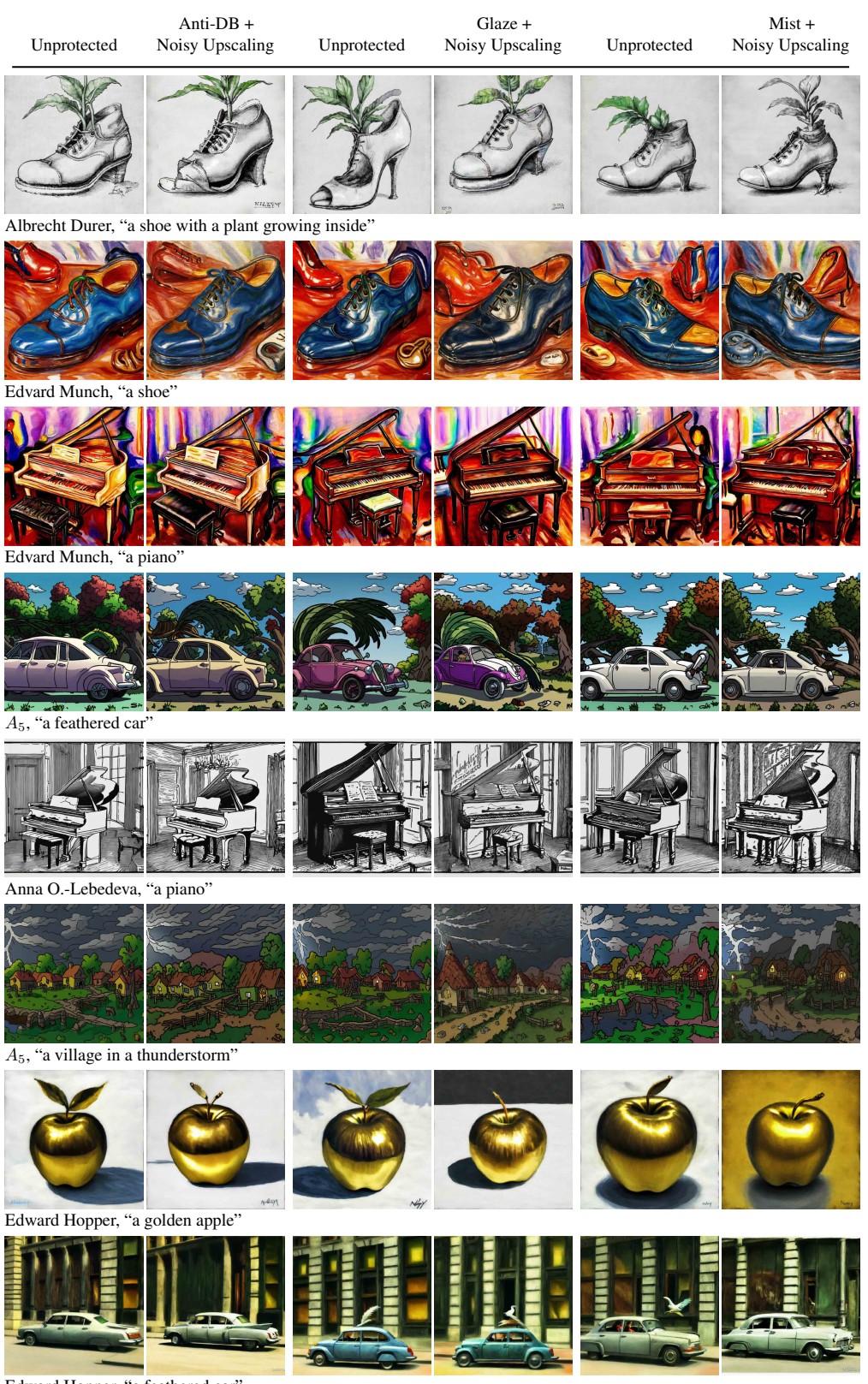

Figure 15: Style mimicry for all protections using *Noisy Upscaling*. We randomly chose artists and prompts. Each image pair shows the protected robust generation and generation from unprotected art.

# C   DETAILED RESULTS

## C.1   MIMICRY QUALITY VERSUS STYLE

This section includes the detailed results from our user study. As mentioned in Section 5, we ask users to assess quality and stylistic fit separately in our study. Figure 16 and 17 show the results for each of these evaluations separately (the results in the main body represent the average of the two). Finally, Table 1 includes numerical results for each scenario.

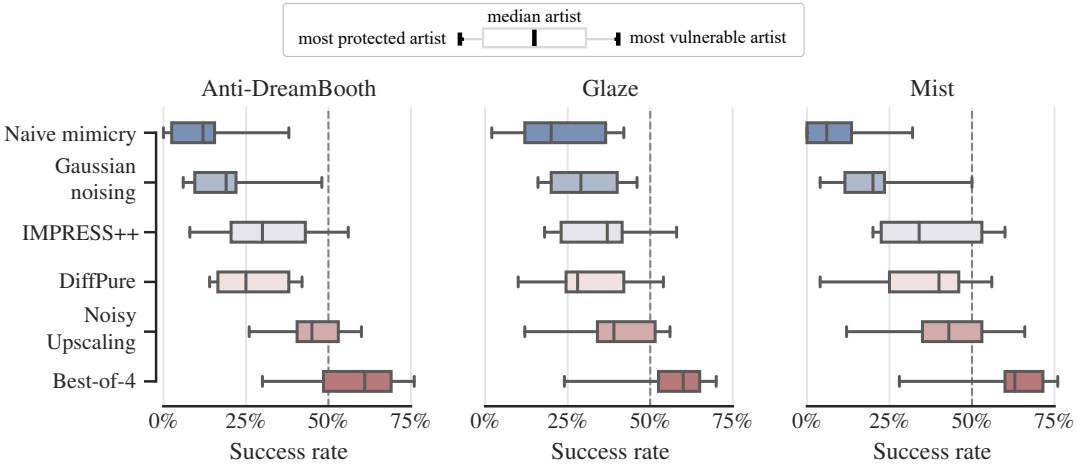

Figure 16: Quality evaluation. User preference ratings of all style mimicry scenarios but only for the quality question: "Based on noise, artifacts, detail, prompt fit, and your impression, which image has higher quality?".

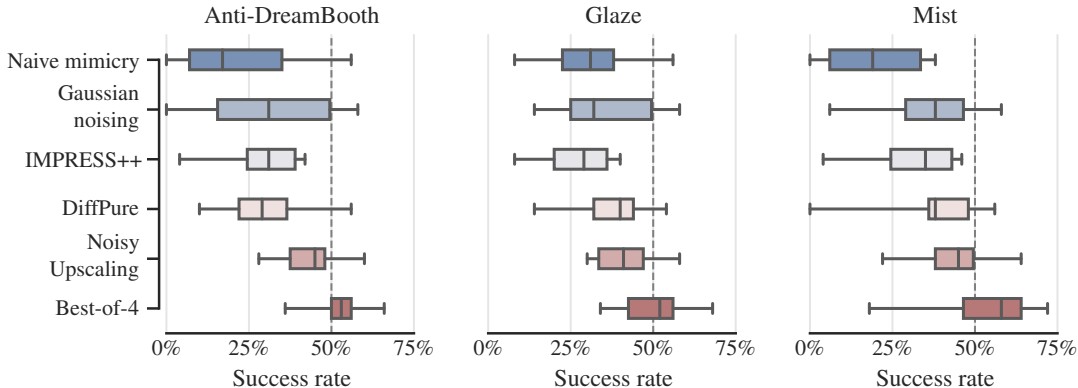

Figure 17: Style evaluation. User preference ratings of all style mimicry scenarios but only for the quality question: "Overall, ignoring quality, which image better fits the style of the style samples?".

Table 1: Success rates averaged across artists for all style mimicry scenarios. Higher percentages indicate more successful mimicry, and 50% would indicate perfect mimicry.

| Method Protection | Naive mimicry | Gaussian noising | IMPRESS++ | DiffPure | Noisy Upscaling | Best-of-4 |
|---|---|---|---|---|---|---|
| Anti-DB | 11.6% | 20.6% | 32.2% | 26.6% | 45.0% | 56.6% |
| Glaze | 22.2% | 29.6% | 35.4% | 32.0% | 39.4% | 56.6% |
| Mist | 9.0% | 21.0% | 37.4% | 35.8% | 42.8% | 62.0% |

(a) Quality

| Method Protection | Naive mimicry | Gaussian noising | IMPRESS++ | DiffPure | Noisy Upscaling | Best-of-4 |
|---|---|---|---|---|---|---|
| Anti-DB | 21.8% | 31.2% | 28.6% | 31.0% | 44.0% | 52.4% |
| Glaze | 30.8% | 35.4% | 27.8% | 37.6% | 41.6% | 51.2% |
| Mist | 19.4% | 35.4% | 31.6% | 37.4% | 44.2% | 53.4% |

(b) Style

## C.2 RESULTS BROKEN DOWN PER ARTIST

We present next the results obtained for each artist in each scenario. Table 2 plots the success rate for each method against each protection for all artists, and Table 3 includes the detailed success rates.

Table 2: Success rates per artist for style and quality questions, respectively. Each line plot shows, for a given protection and artist, the success rate with Gaussian noising ( ▪ ), naive mimicry ( ◇ ), IMPRESS++ ( ● ), DiffPure ( ⋆ ), Noisy Upscaling ( ▲ ), and Best-of-4 ( ✳ ) on a scale from 0% to 77%, where the bar | demarcates 50%.

| Attack Artist | Anti-DB | Glaze | Mist |
|---|---|---|---|
| $A_1$ | | | |
| $A_2$ | | | |
| $A_3$ | | | |
| $A_4$ | | | |
| $A_5$ | | | |
| Albrecht Durer | | | |
| Alphonse Mucha | | | |
| Anna O.-Lebedeva | | | |
| Edvard Munch | | | |
| Edward Hopper | | | |

(a) Quality

| Attack Artist | Anti-DB | Glaze | Mist |
|---|---|---|---|
| $A_1$ | | | |
| $A_2$ | | | |
| $A_3$ | | | |
| $A_4$ | | | |
| $A_5$ | | | |
| Albrecht Durer | | | |
| Alphonse Mucha | | | |
| Anna O.-Lebedeva | | | |
| Edvard Munch | | | |
| Edward Hopper | | | |

(b) Style

Table 3: User preference ratings of all style mimicry scenarios $\mathcal{S} \in \mathbb{M}$ for each artist $A \in \mathbb{A}$ by name. Each cell states the percentage of votes that prefer an image generated under the corresponding scenario $\mathcal{S}$ and artist $A \in \mathbb{A}$ over a matching image generated under clean style mimicry. Higher percentages indicate weaker attacks or better defenses.

| Protection | Method Artist | Naive mimicry | Gaussian noising | IMPRESS++ | DiffPure | Noisy Upscaling | Best-of-4 |
|---|---|---|---|---|---|---|---|
| Anti-DB | $A_1$ | 4% | 6% | 8% | 18% | 26% | 30% |
| | $A_2$ | 14% | 48% | 54% | 32% | 50% | 62% |
| | $A_3$ | 10% | 8% | 18% | 16% | 40% | 46% |
| | $A_4$ | 14% | 22% | 20% | 14% | 54% | 70% |
| | $A_5$ | 16% | 16% | 22% | 24% | 54% | 60% |
| | Albrecht Durer | 2% | 22% | 32% | 26% | 42% | 70% |
| | Alphonse Mucha | 16% | 22% | 44% | 42% | 60% | 66% |
| | Anna O.-Lebedeva | 38% | 40% | 56% | 40% | 44% | 76% |
| | Edvard Munch | 2% | 14% | 40% | 40% | 46% | 56% |
| | Edward Hopper | 0% | 8% | 28% | 14% | 34% | 30% |
| Glaze | $A_1$ | 8% | 20% | 22% | 10% | 12% | 24% |
| | $A_2$ | 12% | 42% | 40% | 28% | 44% | 60% |
| | $A_3$ | 12% | 26% | 18% | 26% | 34% | 52% |
| | $A_4$ | 22% | 20% | 20% | 54% | 54% | 60% |
| | $A_5$ | 18% | 34% | 34% | 24% | 40% | 52% |
| | Albrecht Durer | 2% | 16% | 40% | 28% | 26% | 54% |
| | Alphonse Mucha | 40% | 44% | 58% | 42% | 56% | 66% |
| | Anna O.-Lebedeva | 42% | 46% | 54% | 44% | 34% | 70% |
| | Edvard Munch | 40% | 16% | 42% | 42% | 38% | 62% |
| | Edward Hopper | 26% | 32% | 26% | 22% | 56% | 66% |
| Mist | $A_1$ | 0% | 6% | 20% | 4% | 12% | 28% |
| | $A_2$ | 14% | 50% | 50% | 46% | 48% | 76% |
| | $A_3$ | 0% | 10% | 22% | 24% | 60% | 60% |
| | $A_4$ | 0% | 16% | 24% | 36% | 66% | 70% |
| | $A_5$ | 12% | 22% | 40% | 28% | 50% | 54% |
| | Albrecht Durer | 10% | 24% | 28% | 46% | 38% | 60% |
| | Alphonse Mucha | 32% | 18% | 60% | 56% | 54% | 66% |
| | Anna O.-Lebedeva | 20% | 38% | 54% | 50% | 34% | 74% |
| | Edvard Munch | 2% | 22% | 54% | 44% | 28% | 72% |
| | Edward Hopper | 0% | 4% | 22% | 24% | 38% | 60% |

(a) Quality

| Protection | Method Artist | Naive mimicry | Gaussian noising | IMPRESS++ | DiffPure | Noisy Upscaling | Best-of-4 |
|---|---|---|---|---|---|---|---|
| Anti-DB | $A_1$ | 0% | 4% | 4% | 10% | 34% | 36% |
| | $A_2$ | 14% | 20% | 40% | 16% | 48% | 54% |
| | $A_3$ | 10% | 14% | 26% | 28% | 42% | 46% |
| | $A_4$ | 36% | 58% | 42% | 56% | 54% | 56% |
| | $A_5$ | 4% | 0% | 10% | 32% | 60% | 66% |
| | Albrecht Durer | 20% | 32% | 36% | 28% | 44% | 50% |
| | Alphonse Mucha | 56% | 56% | 42% | 52% | 48% | 58% |
| | Anna O.-Lebedeva | 32% | 50% | 24% | 30% | 28% | 56% |
| | Edvard Munch | 6% | 30% | 26% | 20% | 46% | 50% |
| | Edward Hopper | 40% | 48% | 36% | 38% | 36% | 52% |
| Glaze | $A_1$ | 8% | 14% | 8% | 14% | 30% | 34% |
| | $A_2$ | 36% | 42% | 26% | 46% | 44% | 52% |
| | $A_3$ | 24% | 24% | 16% | 40% | 32% | 50% |
| | $A_4$ | 56% | 58% | 32% | 44% | 58% | 66% |
| | $A_5$ | 12% | 18% | 18% | 30% | 32% | 40% |
| | Albrecht Durer | 22% | 28% | 26% | 26% | 38% | 38% |
| | Alphonse Mucha | 48% | 54% | 36% | 54% | 52% | 56% |
| | Anna O.-Lebedeva | 26% | 32% | 40% | 38% | 44% | 68% |
| | Edvard Munch | 38% | 32% | 36% | 40% | 48% | 56% |
| | Edward Hopper | 38% | 52% | 40% | 44% | 38% | 52% |
| Mist | $A_1$ | 0% | 6% | 4% | 0% | 22% | 18% |
| | $A_2$ | 6% | 38% | 44% | 42% | 64% | 72% |
| | $A_3$ | 6% | 28% | 26% | 36% | 34% | 44% |
| | $A_4$ | 36% | 58% | 46% | 52% | 48% | 54% |
| | $A_5$ | 4% | 14% | 18% | 26% | 58% | 56% |
| | Albrecht Durer | 28% | 32% | 24% | 36% | 50% | 60% |
| | Alphonse Mucha | 34% | 50% | 34% | 50% | 48% | 64% |
| | Anna O.-Lebedeva | 32% | 48% | 44% | 56% | 38% | 64% |
| | Edvard Munch | 10% | 38% | 36% | 40% | 42% | 64% |
| | Edward Hopper | 38% | 42% | 40% | 36% | 38% | 38% |

(b) Style

## C.3 INTER-ANNOTATOR AGREEMENT

**Inter-annotator agreement**

Robust mimicry

Naive mimicry

0%  20%  40%  60%  80%  100%

■ 3/5 votes agree   ■ 4/5 votes agree   ■ 5/5 votes agree

Figure 18: Inter-annotator agreement for generations from robust mimicry with Noisy Upscaling and generations from models finetuned on protected art directly (naive mimicry). We plot the percentage of comparisons for which the preferred option was selected by 3, 4 or 5 annotators, respectively. The graph shows a higher consensus for naive mimicry, since the differences are clearer, and more variance for robust mimicry.

## D DIFFERENCES WITH GLAZE FINETUNING

In Section 4.1 and Figure 2, we discussed the brittleness of Glaze protections against small changes in the finetuning script. We also found our finetuning setup to be better at baseline style mimicry from unprotected art (see Figure 19).

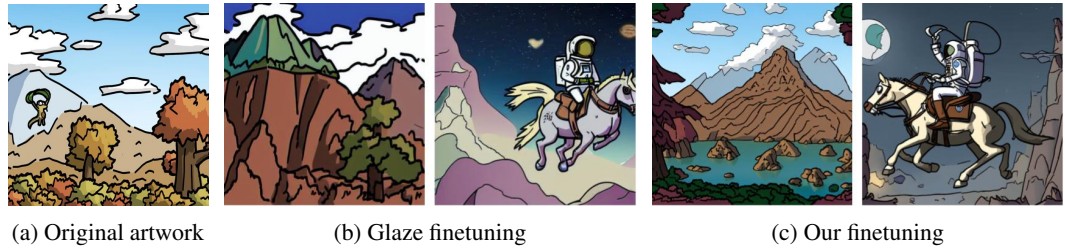

(a) Original artwork        (b) Glaze finetuning        (c) Our finetuning

Figure 19: The finetuning script shared by Glaze authors produce substantially worse mimicry even from unprotected art. We apply both finetuning scripts directly on unprotected art from @nulevoy. The main reason behind this difference might be that the script uses Stable Diffusion 1.5, instead of version 2.1 as reported in their paper.

# E    FINDINGS ON GLAZE 2.0

After concluding our user study, Glaze (Shan et al., 2023a) released an updated version of their tool (v2.0). According to the official release, "This new version significantly improved Glaze robustness against the newest AI models". Although we could not run the entire user study with the latest protections, we reproduced some of our experiments to verify if protections were more robust under robust mimicry. We believe this comparison is fair to Glaze since we are using newer models—such as Stable Diffusion XL for upscaling. These models, although released before Glaze 1.1.1, may not have been considered in the tool's design and are now explicitly accounted for.

The official release specifically mentions "Significantly improved robustness against Stable Diffusion 1, 2, SDXL, especially for smooth surface art (e.g. anime, cartoon)". Therefore, we decided to test this new tool with the contemporary artist *nulevoy*, who draws in a cartoon style and gave us permission to display their artwork. As with the previous version, we only have access to the publicly available Windows application that uses unknown parameters. We protect the images using the "highest" protection option. Our main findings are:

1. Glaze v2.0 introduces more visible perturbations uniformly over the images. See Figure 20.

2. Glaze v2.0 does not improve protection under robust mimicry. Noisy Upscaling still achieves almost perfect style mimicry. See Figure 21.

3. Noisy Upscaling is able to to remove visible perturbations during preprocessing as before. See Figure 22.

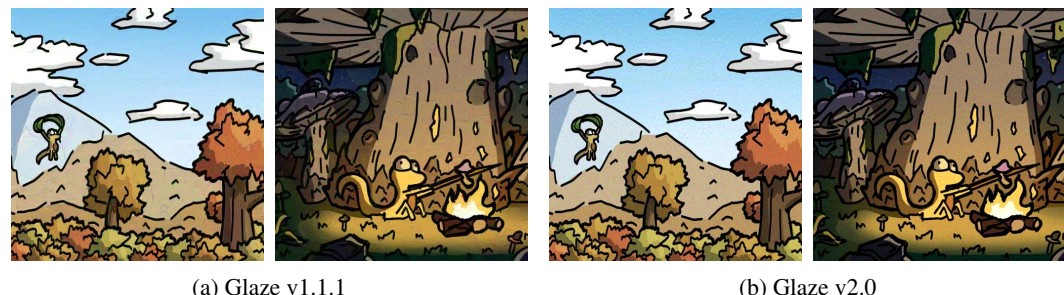

(a) Glaze v1.1.1                                         (b) Glaze v2.0

Figure 20: Comparison of perturbations by Glaze v1.1.1 and v2.0 on artwork from @*nulevoy*.

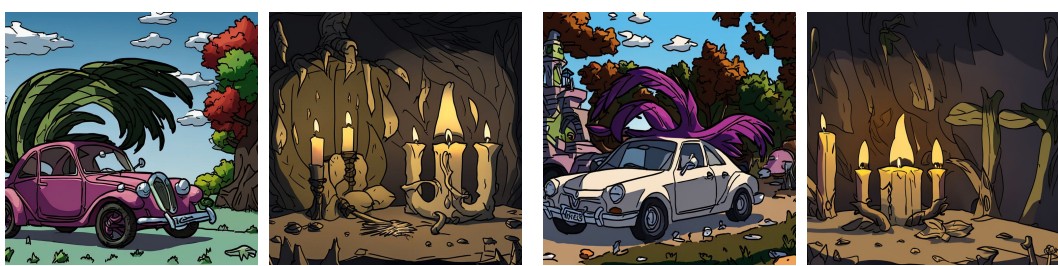

(a) Robust style mimicry on Glaze v1.1.1          (b) Robust style mimicry on Glaze v2.0

Figure 21: Comparison of robust style mimicry (Noisy Upscaling) on artwork from @*nulevoy* protected with both versions of Glaze. Images in Figure 6 serve as a reference for the artistic style.

# F    FINDINGS ON MIST V2

After responsibly disclosing our work to defense developers, authors from Mist brought to our attention the recent release of their latest Mist v2 with improved resilience (Zheng et al., 2023). As we did with Glaze v2.0 (see Section E), we reproduced some of our experiments with the latest

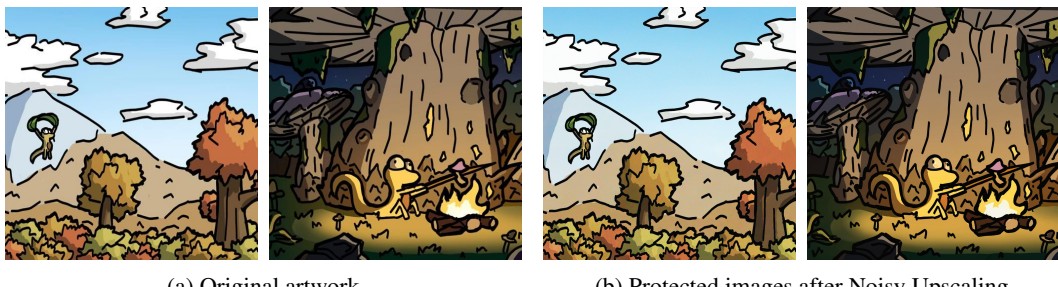

(a) Original artwork      (b) Protected images after Noisy Upscaling

Figure 22: Original artwork from *@nulevoy* and the resulting images after applying Noisy Upscaling to artwork protected with Glaze v2.0. See protected images in Figure 20.

protections to verify the success of robust mimicry. Their original implementation still uses the outdated version 1.5 of Stable Diffusion. We change to SD 2.1 to match our previous experiments[7].

Our findings, as we saw with Glaze v2.0, highlight that improved protections are still not effective against low-effort robust mimicry. More specifically, the latest version of Mist:

1. introduces visible perturbations over the images. See Figure 23
2. does not improve protections against robust mimicry. See Figure 24
3. creates protection that are easily removable with Noisy Upscaling. See Figure 25.

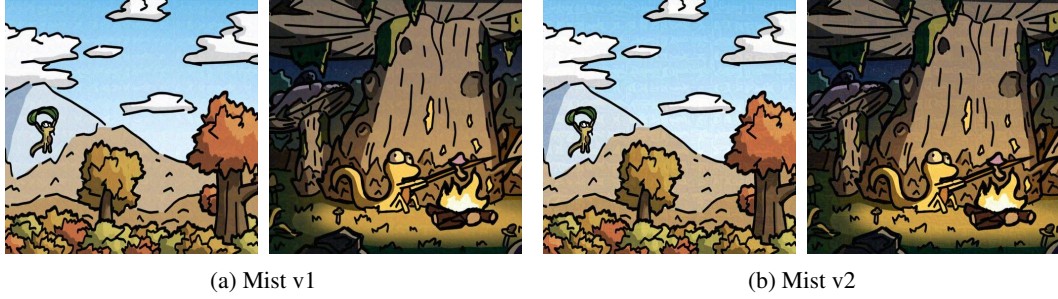

(a) Mist v1      (b) Mist v2

Figure 23: Comparison of perturbations introduced by Mist v1 and v2 on artwork from *@nulevoy*.

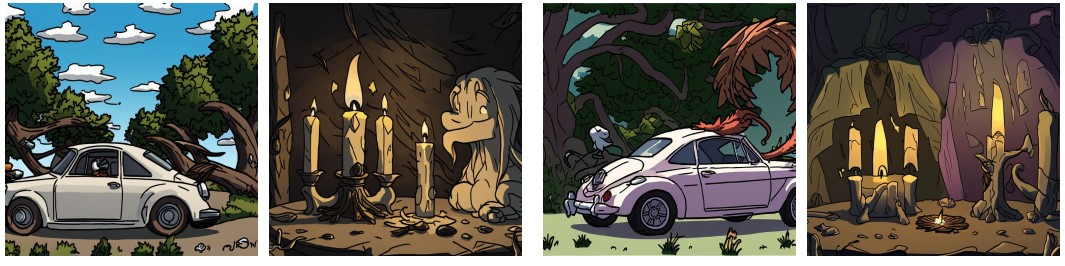

(a) Robust style mimicry on Mist v1      (b) Robust style mimicry on Mist v2

Figure 24: Comparison of robust style mimicry (Noisy Upscaling) on artwork from *@nulevoy* protected with both versions of Mist. Images in Figure 6 serve as a reference for the artistic style.

---

[7]Both models share the same encoder for which protections are optimized.

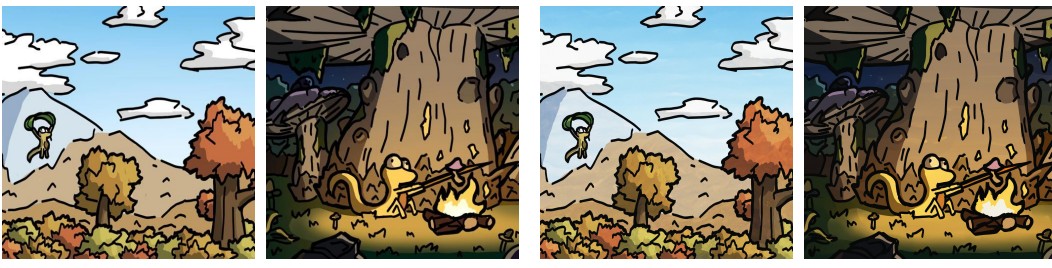

(a) Original artwork          (b) Protected images after Noisy Upscaling

Figure 25: Original artwork from *@nulevoy* and the resulting images after applying Noisy Upscaling to artwork protected with Mist v2. See protected images in Figure 23.

## G    METHODS FOR STYLE MIMICRY

This section summarizes the existing methods that a style forger can use to perform style mimicry. Our work only considers *finetuning* since it is reported to be the most effective (Shan et al., 2023a).

### G.1    PROMPTING

Well-known artistic styles contained in the training data (e.g. Van Gogh) can be mimicked by prompting a text-to-image model with a description of the style or the name of the artist. For example, a prompt can be augmented with " painted in a cubistic style" " painted by van Gogh" to mimic those styles, respectively. Prompting is easy to apply and does not require changes to the model. However, it fails to mimic styles that are not sufficiently represented in the training data of model—often from the most vulnerable artists.

### G.2    IMG2IMG

Img2Img creates an updated version of an image with guidance from a prompt. For this, Img2Img processes image $x$ with $t$ timesteps of a diffusion process to obtain the diffused image $x_t$. Then, Img2Img uses the model with guidance from prompt $P$ to reverse the diffusion process into the output image variation $x_P$. Analogous to prompting, a prompt suffices to transfer a well-known style, but Img2Img also fails for unknown styles.

### G.3    TEXTUAL INVERSION

Textual inversion (Gal et al., 2022) optimizes the embedding of some $n$ new tokens $\boldsymbol{t} = [t_1, \ldots, t_n]$ that are appended to image prompts $P$ so that generations closely mimic the style of a given set of images. The tokens are optimized via gradient descent on the model training loss so that $P + \boldsymbol{t}$ generates images that mimic the target style. Textual inversion requires white-box access to the target model, but enables the mimicry of unknown styles.

### G.4    FINETUNING

Finetuning updates the weights of a pretrained text-to-image model to introduce a new functionality. In this case, finetuning allows a forger to "teach" the generative model an unknown style using a set of images in the target style and their captions (e.g. *an astronaut riding a horse*). First, all captions are augmented with some special word, like the name of the artist, to create prompts $P_x = C_x +$ "by $w_*$". Then, the model weights are updated to minimize the reconstruction loss of the given images following the augmented prompts. At inference time, the forger can append "by $w_*$" to any prompt to obtain art in the target style

The authors of Glaze identify this finetuning setup as the strongest style mimicry method (Shan et al., 2023a). We validate the success of our style mimicry with a user study detailed in Appendix K.1

## H    EXISTING STYLE MIMICRY PROTECTIONS

**Naming convention.**    Depending on the context, style mimicry protections may be viewed either as attacks or as the targets of attacks. In an artistic setting, artists see style mimicry as an attack and utilize methods like Glaze as a defense. Conversely, in the context of adversarial robustness, Glaze can be seen as an attack against style mimicry methods through adversarial perturbations. The research community has not reached a consensus on terminology: Glaze's authors consider style mimicry an attack and label Glaze as a defense, while the authors of Mist and Anti-DreamBooth describe their approaches as attacks. In our work, we distance ourselves from the attack/defense terminology and instead refer to these mechanisms as protections, and to the party performing mimicry as the "style forger".

Existing protections can either target the encoder or the decoder of text-to-image models. We classify them accordingly.

### H.1    ENCODER PROTECTIONS

Encoder protections include adversarial perturbations in the images $X$ so that the encoder $\mathcal{E}_{\phi}$ of the model maps images to latent representations that, when reconstructed, recover images in a different style. Concretely, an encoder protection first defines a target latent representation $t_x \in \text{Latent}$ for each image $x \in X$ that is different to its own style. For instance, the target latent representation for Edvard Munch could be Vincent Van Gogh. Then, protection $\mathcal{P}$ optimizes the objective

$$\min_{\boldsymbol{\delta}_x} \mathrm{d}_{\text{Lat}}(\mathcal{E}_{\phi}(x + \boldsymbol{\delta}_x), t_x)$$
$$\text{subject to} \quad \mathrm{d}_{\text{Img}}(x + \boldsymbol{\delta}_x, x) \leq p. \tag{2}$$

**Glaze** (Shan et al., 2023a) is an instance of an encoder protection. Glaze first selects an adversarial target style $\mathcal{S}_{\text{adv}}$ that style mimicry should learn instead of the style $\mathcal{S}$ to be protected. Then, Glaze uses Img2Img style transfer to create a variation $x_{\mathcal{S}_{\text{adv}}}$ in style $\mathcal{S}_{\text{adv}}$ of each image $x \in X$. The latent representation of variation $x_{\mathcal{S}_{\text{adv}}}$ is used as the target latent representation $t_x$ for each image $x \in X$.

Glaze selects the target style $\mathcal{S}_{\text{adv}}$ from a pre-defined set of 50 styles $\mathbb{S}_{\text{adv}}$. First, Glaze computes the distance between the mean CLIP embedding of the images $X$ and the prompt $P_{S'}$ corresponding to each style $S' \in \mathbb{S}_{\text{adv}}$. Then, Glaze randomly samples target style $\mathcal{S}_{\text{adv}}$ from the 50 to the 75 percentile of target styles $\mathbb{S}_{\text{adv}}$ sorted by distance.

Glaze implements Objective (2) with the penalty method (Wright, 2006) as

$$\min_{\boldsymbol{\delta}_x} \|\mathcal{E}_{\phi}(x + \boldsymbol{\delta}_x), t_x\|_2^2 + \alpha \cdot \max(\text{LPIPS}(x + \boldsymbol{\delta}_x, x) - p, 0) \tag{3}$$

where LPIPS (Zhang et al., 2018) is a choice for metric $d_{\text{Img}}$ that aims to measure user-perceived image distortion. Glaze then optimizes Objective (3) with the Adam (Kingma & Ba, 2014) optimizer.

**Mist$_{\phi}$** (Liang et al., 2023) is a different encoder protection from the Mist project[8]. Mist$_{\phi}$ optimizes perturbations with PGD to minimize the squared $L_2$-induced distance between the latent representation of the artists' images and some unrelated target image.

In their original work, Mist is only evaluated against DreamBooth, Style Transfer, and Textual Inversion, but not against finetuning. Also, the original Mist work refers to Mist$_{\phi}$ as Mist operating in *textural mode*.

### H.2    DENOISER PROTECTIONS

Denoiser protections use the prediction error of the denoiser $\epsilon_{\boldsymbol{\theta}}$ as a proxy of the quality of style mimicry, making it a feasible target for adversarial optimization. Current Denoiser protections, such as Mist (Liang et al., 2023) and Anti-DreamBooth (Van Le et al., 2023) assume that poorly reconstructed images will fail to mimic style

---

[8]Mist project also contains a denoiser attack that we fail to reproduce as a robust protection.

**Anti-DreamBooth** (Van Le et al., 2023) uses the prediction error of the denoiser $\epsilon_{\boldsymbol{\theta}_{\mathrm{adv}}}$ as a proxy for the mimicry quality, where denoiser $\epsilon_{\boldsymbol{\theta}_{\mathrm{adv}}}$ corresponds to the denoiser from a finetuned model trained on images with the style to be protected. Since perturbations maximizing the error with the pretrained decoder can be easily circumvented with finetuning, Anti-DreamBooth uses a technique they refer to as *Alternating Surrogate and Perturbation Learning* (ASPL). The intuition behind ASPL is trying to simulate finetuning on the art and maximizing the error during finetuning. For this purpose, they interleave finetuning steps with perturbation optimization steps.

## I   ROBUST MIMICRY METHODS

This section details the robust mimicry methods we use in our work. These methods are not aimed at maximizing performance. Instead, they demonstrate how various "off-the-shelf" and low-effort techniques can significantly weaken style mimicry protections.

Formally, given protected images $X$ and a pretrained text-to-image model $f$, we define a general robust mimicry pipeline that finetunes a model $\hat{f}$ and then produces an image $Z$ for a given *prompt* as follows (a successful method may not require modifications in all stages):

$$\hat{f} \leftarrow \texttt{Finetune}(f; \texttt{PreProcess}(X))$$
$$Z \leftarrow \texttt{PostProcess}(\texttt{Sample}(\hat{f}, \text{``prompt''})).$$

### I.1   DIFFPURE

DiffPure (Nie et al., 2022) uses image generation diffusion models to adversarially purify images $X_{\mathrm{prot}}$. DiffPure processes each image $x_{\mathrm{adv}} \in X_{\mathrm{prot}}$ with $t$ timesteps of a diffusion process to obtain the diffused image $x_{\mathrm{adv}}^t = \sqrt{\alpha_t} \cdot x_{\mathrm{adv}} + \sqrt{1 - \alpha_t} \cdot \epsilon$, where $\alpha$ is the noise schedule of the diffusion process and noise $\epsilon$ is sampled from $\mathcal{N}(0, \boldsymbol{I})$. Then, DiffPure constructs the purified image $\mathrm{DiffPure}(x_{\mathrm{adv}})$ by applying reverse diffusion to image $x_{\mathrm{adv}}^t$ for $t$ timesteps with an image generation diffusion model DM. Nie et al. prove that under certain idealized conditions, DiffPure is likely to weaken adversarial perturbations in image $x_{\mathrm{adv}}$.

If the text-to-image model M supports unconditional image generation, then we can use model M for the reverse diffusion process. For example, Stable Diffusion (Rombach et al., 2022) generates images unconditionally when the prompt $P$ equals the empty string. Under these conditions, Img2Img is equivalent to DiffPure. Therefore, in the context of defenses for style mimicry, we refer to Img2Img applied with an empty prompt $P$ as *unconditional DiffPure*, and to Img2Img applied with a non-empty prompt $P$ as *conditional DiffPure*.

### I.2   NOISY UPSCALING

Upscaling increases the resolution of an image by predicting new pixels that enhance the level of detail. Upscaling images can purify adversarially perturbed images (Mustafa et al., 2019). However, we discover that applying upscaling directly on protected images fails to remove the perturbations.

We define *Noisy Upscaling* as a way to address the shortcomings of upscaling. Noisy Upscaling first applies Gaussian noising and then upscales the noisy image. Noisy Upscaling has a more profound effect than the sum of its parts: Gaussian noising only adds noise to an image $x_{\mathrm{adv}}$, but does not remove the adversarial perturbation $\boldsymbol{\delta}_x$. Similarly, we observe upscaling to roughly preserve perturbation $\boldsymbol{\delta}_x$. In contrast, $\mathrm{NoisyUpscale}(x_{\mathrm{adv}})$ shows neither visually perceptible noise, nor adversarial perturbations. Figure 26 illustrates the improvements. We explain these phenomena as follows.

First, we use the Stable Diffusion Upscaler ($\mathrm{Upscale}_{\mathrm{SD}}$), which is trained on noise-augmented images and accepts the corresponding noise level $L$ as a class-conditioning label. We can therefore condition $\mathrm{Upscale}_{\mathrm{SD}}$ on the noise level $L_{\sigma^2}$, corresponding to the variance $\sigma^2$ used by GaussianNoising, to remove the noise that GaussianNoising adds.

Second, we note that upscaling has shown success against adversarial perturbations for classifiers (Mustafa et al., 2019), but not against adversarial perturbations for generative models (Liang et al., 2023; Shan et al., 2023a).

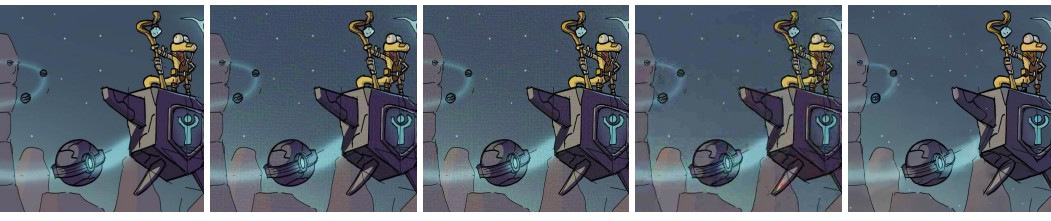

(a) Original artwork  (b) Protected artwork  (c) Upscaling  (d) Compr. Upscaling  (e) *Noisy Upscaling*

Figure 26: Illustration of Noisy Upscaling on a random image from @nulevoy. Unlike naive upscaling and Compressed Upscaling, Noisy Upscaling removes protections while preserving the details in the original artwork.

## I.3   IMPRESS++

We enhance the IMPRESS algorithm (Cao et al., 2024). We change the loss of the reverse encoding optimization from patch similarity to $l_\infty$ and include two additional steps: negative prompting and post-processing. All in all, IMPRESS++ first preprocesses protected images with Gaussian noise and reverse encoder optimization, then samples using negative prompting and finally post-processes the generated images with DiffPure to remove noise.

**Reverse encoder optimization.** *Reverse encoder optimization* is a preprocessing defense against encoder protections. It adds additional perturbations $\boldsymbol{\Delta}'$ to images $X_{\text{prot}}$ so that the latent representation $\boldsymbol{t}_{x'_{\text{adv}}} = \mathcal{E}_\phi(x'_{\text{adv}})$ of each protected image $x'_{\text{adv}} = x_{\text{adv}} + \boldsymbol{\delta}_{x_{\text{adv}}}$ satisfies

$$\mathcal{D}_{\phi'}\left(\boldsymbol{t}_{x'_{\text{adv}}}\right) \approx x'_{\text{adv}} \tag{4}$$

and each perturbation $\boldsymbol{\delta}_{x_{\text{adv}}} \in \boldsymbol{\Delta}'$ satisfies

$$\mathrm{d}_{\text{Img}}(x_{\text{adv}} + \boldsymbol{\delta}_{x_{\text{adv}}}, x_{\text{adv}}) \le p. \tag{5}$$

If Equation (4) holds, then style mimicry finetuning learns the style of images $X'_{\text{prot}}$. In addition, the combination of Equation (5) with the image similarity constraint $\mathrm{d}_{\text{Img}}(x + \boldsymbol{\delta}_x, x) \le p$ in Objective (2) ensures that the defended images $X'_{\text{prot}}$ look similar to the original images $X$. Therefore, style mimicry finetuning on images $X'_{\text{prot}}$ should learn a style similar to style $\mathcal{S}$.

Reverse encoder optimization aims to achieve Equation (4) and Equation (5) by optimizing the objective

$$\min_{\boldsymbol{\delta}_{x_{\text{adv}}}} \mathrm{d}_{\text{Lat}}(\mathcal{E}_\phi(x_{\text{adv}} + \boldsymbol{\delta}_{x_{\text{adv}}}), \mathcal{E}_\phi(x_{\text{adv}}))$$
$$\text{subject to}\quad \mathrm{d}_{\text{Img}}(x_{\text{adv}} + \boldsymbol{\delta}_{x_{\text{adv}}}, x_{\text{adv}}) \le p \tag{6}$$

with PGD.

**Negative prompting.** Negative prompting (Miyake et al., 2023) is a technique to guide image generation of a diffusion-based text-to-image model M away from a prompt $P_{\text{neg}}$. To this end, negative prompting manipulates the classifier-free guidance (Ho & Salimans, 2022), which computes the denoiser output of model M as

$$\tilde{\epsilon}_{\boldsymbol{\theta}}(\boldsymbol{z}, t, P) = (1 + w) \cdot \epsilon_{\boldsymbol{\theta}}(\boldsymbol{z}, t, P) - w \cdot \epsilon_{\boldsymbol{\theta}}(\boldsymbol{z}, t, \text{``''}) \tag{7}$$

where parameter $w$ controls the guidance strength. Negative prompting simply substitutes the empty string "" with $P_{\text{neg}}$ to obtain

$$\tilde{\epsilon}_{\boldsymbol{\theta}}(\boldsymbol{z}, t, P) = (1 + w) \cdot \epsilon_{\boldsymbol{\theta}}(\boldsymbol{z}, t, P) - w \cdot \epsilon_{\boldsymbol{\theta}}(\boldsymbol{z}, t, P_{\text{neg}}). \tag{8}$$

We design a routine for $\mathcal{D}_{\text{InF}}$ that leverages negative prompting to guide model M away from adversarial generations. To this end, we first apply Textual Inversion with adversarial images $X_{\text{prot}}$ to encode the style of adversarial generations $\mathcal{S}_{\text{adv}}$ into a special word $w_*$. We then set prompt $P_{\text{neg}} =$ "art by $w_*$".

Naive negative prompting offers no strength control. Too little strength may fail to guide model M away from the adversarial style $\mathcal{S}_{\text{adv}}$. Too much strength may guide towards the style opposite to style $\mathcal{S}_{\text{adv}}$ in the latent space of model M, which is not necessarily the desired style $\mathcal{S}$. We use negative prompt weights (muerrilla, 2023) to control the strength of negative prompting. The negative prompt weights technique introduces the strength control parameter $c$ to interpolate between Equation (7) and Equation (8) as

$$\tilde{\epsilon}_{\boldsymbol{\theta}}(\boldsymbol{z}, t, P) = (1 + w) \cdot \epsilon_{\boldsymbol{\theta}}(\boldsymbol{z}, t, P) - w \cdot ((1 + c) \cdot \epsilon_{\boldsymbol{\theta}}(\boldsymbol{z}, t, P_{\text{neg}}) - c \cdot \epsilon_{\boldsymbol{\theta}}(\boldsymbol{z}, t, \text{""})). \quad (9)$$

Figure 27 illustrates the improvements introduced by each additional step.

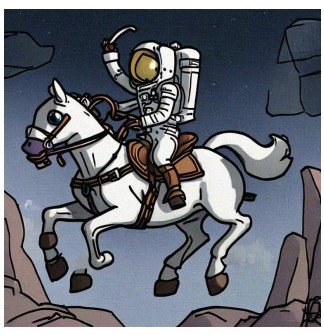 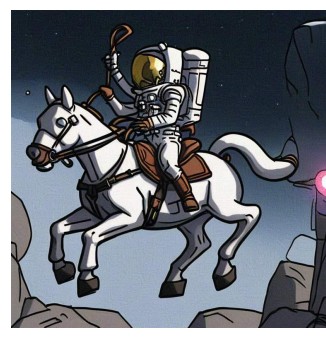 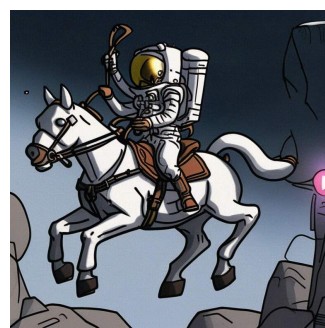

|  |  |  |
|---|---|---|
| (a) Original IMPRESS | (b) IMPRESS + negative prompting | (c) *IMPRESS++*. IMPRESS + negative prompting + denoising |

Figure 27: Improvements of each additional step in IMPRESS++ over the original IMPRESS (Cao et al., 2024). Negative prompting improves image consistency and denoising reduces artifacts in generated images.

## J   EXPERIMENTAL SETUP

This section describes our general experimental setup and specifies the settings and hyperparameters of the methods we use. When possible, we use default values from the machine learning literature. For implementation details see our official repository: `https://github.com/ethz-spylab/robust-style-mimicry`

### J.1   STYLE MIMICRY EXPERIMENTAL DETAILS

As described in Section 3, our threat model considers style mimicry with a latent diffusion text-to-image model M that is finetuned on a set of images $X$ in a style $\mathcal{S}$. This section specifies our choices for model M, images $X$, style $\mathcal{S}$, the hyperparameters for finetuning M, and the hyperparameters for generating images with the finetuned model. Where possible, we try to replicate the style mimicry setup used by Shan et al. to evaluate Glaze, and highlight any differences.

**Model**   We use Stable Diffusion version 2.1 (Stability AI, 2022), the same model used to optimize the protections we evaluate (Shan et al., 2023a; Liang et al., 2023; Van Le et al., 2023).

**Dataset.**   We collate 10 image sets $\left\{X^A : A \in \mathbb{A}\right\}$ from 10 different artists $\mathbb{A}$. Each image set $X^A$ contains 18 images that we choose manually to follow a consistent style $\mathcal{S}_A$. We select the artists $\mathbb{A}$ from contemporary and historical artists: We select 5 contemporary artists from ArtStation[9] and 5 historical artists from the WikiArt dataset (Tan et al., 2019). We found 2 of the 4 artists used by Glaze and included them in our evaluation. We manually select the remaining 8 artists to cover a broad variety of styles. Glaze additionally verified that the images of the contemporary artists in their evaluation are not included in the training dataset of the model M. Unfortunately, the LAION-5B dataset (Schuhmann et al., 2022) used to train SD 2.1 was taken offline (Cole, 2023), so we are unable to perform this verification. Instead, we verify for each contemporary artist $A \in \mathbb{A}$ that SD 2.1 is

---

[9]`www.artstation.com`

unable to mimic the style $\mathcal{S}_A$ by manually inspecting SD 2.1 generations for prompts of the form "An {object} by {artist}". We center-crop each image $x$ to $512 \times 512$ pixels and generate a caption $C_x$ for $x$ with the BLIP-2 model (Li et al., 2023).

**Finetuning hyperparameters.** Glaze does not specify which finetuning script they use, but they claim to "follow the same training parameters as (Rombach et al., 2022). We use $5 \cdot 10^{-6}$ learning rate and batch size of 32." This batch size misfits their small finetuning image sets that contain no more than 34 images. Moreover, the finetuning code that Shan et al. kindly sent us upon request uses DreamBooth finetuning with Stable Diffusion 1.5, instead of version 2.1 as described in their work.

In light of these discrepancies, and assuming that mimicry protections should be agnostic to the finetuning setup used, we use an "off-the-shelf" HuggingFace finetuning script for Stable Diffusion (von Platen et al., 2024) and manually tune hyperparameters for optimal style mimicry before protections are applied. Concretely, we use 2,000 training steps, batch size 4, learning rate $5 \cdot 10^{-6}$, and set the remaining hyperparameters to their default values. We pair each image $x$ with the prompt $P_x = C_x +$" by $w_*$", where $w_* = $ "nulevoy"[10].

**Generation hyperparameters** We use the DPM-Solver++(2M) Karras (Lu et al., 2022; Karras et al., 2022) scheduler for 50 steps to generate images of size $768 \times 768$. This scheduler generates images with slightly higher quality than the PNDM (Liu et al., 2021) scheduler used by Glaze.

### J.2 PROTECTIONS EXPERIMENTAL DETAILS

We evaluate three different protections: Mist (Liang et al., 2023), Glaze (Shan et al., 2023a), and Anti-DreamBooth (Van Le et al., 2023). For a fair comparison, we fix the perturbation budget $p$ for each adversarial perturbation $\boldsymbol{\delta}_x$ created by Mist and Anti-DreamBooth to $p = 8/255$, which is the same budget that Liang et al. use to evaluate Mist. It is not possible to evaluate Glaze with exactly this perturbation budget, for three reasons: First, Glaze uses LPIPS for the image similarity measure $d_{\mathrm{Img}}$, which does not bound the $L_\infty$ norm. Second, Glaze implements the metric $d_{\mathrm{Img}}$ as a soft bound in Objective (3), which offers no hard bound guarantees. Third, Glaze is closed-source software whose perturbation budget control only offers the settings `Default`, `Medium`, and `High`. Upon request, the Glaze authors refused to share a codebase where we could control the hyperparameters. Therefore, we evaluate Glaze through their official public tool with the setting `High` to evaluate our protections under the highest protections. In our evaluation, we perceive images processed with Glaze to be equally or less perturbed than images processed with Mist and Anti-DreamBooth.

Next, we describe specific hyperparameters we use to reproduce each of the protections.

#### J.2.1 ANTI-DREAMBOOTH

Van Le et al. implement Anti-DreamBooth against DreamBooth finetuning. We adapt their implementation to our vanilla finetuning for style mimicry, using the same hyperparameters where possible: We set the number of iterations to $N = 50$, the PGD perturbation budget to $p = 8/255$, the PGD step size to $\alpha = 5 \cdot 10^{-3}$, and the number of PGD steps per ASPL iteration to $N_{\mathrm{PGD}} = 6$. We minimize the loss $\mathcal{L}_{\mathrm{Finetune}}$ with the vanilla finetuning setup in Appendix J.1 for 300 training steps.

#### J.2.2 MIST$_\phi$

We replicate the evaluation that Liang & Wu use to evaluate Mist$_\phi$ against Stable Diffusion. We set the PGD perturbation budget to $p = 8/255$, the number of PGD iterations to $N_{\mathrm{PGD}} = 100$, the PGD step size to $\alpha = 1/255$, and the target image to $T = \mathrm{Target\_Mist}$ shown in Figure 28.

#### J.2.3 GLAZE

The Glaze authors were unable to share a codebase upon request. We thus use their publicly released Windows application binary. We use the latest available version of Glaze, v1.1.1. We set `Intensity`

---

[10]@nulevoy is the first ArtStation artist that we experimented with. In our experiments, we found "nulevoy" a suitable choice for the special word $w_*$ and use it for all artists. We check that all of nulevoy's images are published after the release date of LAION-5B to ensure that SD 2.1 has no prior knowledge about nulevoy's style.

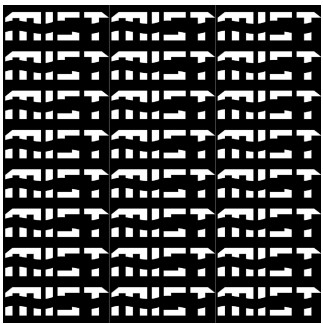

Figure 28: The Mist target image Target_Mist. Target_Mist is the default target image in the reference Mist implementation and one of the successful target images evaluated by Liang & Wu.

to `High` and `Render Quality` to `Slowest`, to obtain the strongest protections. Appendix E includes qualitative results on an updated version released after we concluded our user study.

### J.3 Robust Mimicry Methods Experimental Details

#### J.3.1 Gaussian Noising

We manually tune the Gaussian noising strength to $\sigma_2 = 0.05$.

#### J.3.2 DiffPure

We use conditional DiffPure with the best-performing publicly available image generation diffusion model, Stable Diffusion XL 1.0 (SDXL) (Podell et al., 2023). We implement conditional DiffPure using the HuggingFace `AutoPipelineForImage2Image` pipeline. We use classifier-free guidance scale `guidance_scale = 7.5` with prompt $P = C_x$ for image $x$. We manually tune the number of diffusion timesteps $t$ via the `strength` pipeline argument to `strength = 0.2`.

#### J.3.3 IMPRESS++

**Reverse Optimization** Like Mist$_\phi$, we set the PGD perturbation budget to $p = 8/255$ and the PGD step size to $\alpha = 1/255$. We manually tune the number of PGD iterations to $N_{\text{PGD}} = 400$.

**Noisy Upscaling** We manually tune the Gaussian noising strength to $\sigma = 0.1$. We then use the Stable Diffusion Upscaler [11] with the maximum denoising strength $L$.[12]

We note that the Stable Diffusion Upscaler is trained on diffused images of the form $x_\alpha = \sqrt{\alpha} \cdot x + \sqrt{1-\alpha} \cdot \mathcal{N}(0, \boldsymbol{I})$. In contrast, noisy upscaling noises images additively, that is, without the factor $\sqrt{\alpha}$. However, we note that for $\sqrt{1-\alpha} = \sigma = 0.1$, we have $\sqrt{\alpha} = 0.995 \approx 1$. In practice, we observe no qualitative difference in the generated images.

**Negative Prompting** We manually tune the negative prompting strength to $c = 0.5$. We use the Stable Diffusion web UI [13] to apply Textual Inversion on the adversarial images $X_{\text{prot}}$. We follow the Textual Inversion setup used by Liang et al. to evaluate Mist and set the length of the token vector $\boldsymbol{t}$ to $n = 8$, the embedding initialization text to "style *", the learning rate to $\gamma = 0.005$, the batch size to 1, and the number of training steps to 500.

**DiffPure$_{\text{post}}$** To make IMPRESS++ work under a single-model availability, we apply DiffPure$_{\text{post}}$ with the same model that we use for image generation, SD 2.1. We implement DiffPure$_{\text{post}}$ using the HuggingFace `AutoPipelineForImage2Image` pipeline. We use the classifier-free guidance

---

[11] www.huggingface.co/stabilityai/stable-diffusion-x4-upscaler

[12] We inadvertently set the denoising strength to $L = 320$ instead of the actual maximum denoising strength $L = 350$. We observe no qualitative difference in the generated images.

[13] https://github.com/AUTOMATIC1111/stable-diffusion-webui

scale `guidance_scale = 7.5` with prompt $P = C_x +$ ", artistic" for image $x$. We manually tune the number of diffusion timesteps $t$ via the `strength` pipeline argument to the value `strength =` 0.2.

## K  USER STUDY

This user study was approved by our institution's IRB.

**Design.** Our user study asks annotators to compare outputs from one robust mimicry method against a baseline where images are generated from a model trained on the original art without protections—for a fixed set of prompts $\mathbb{P}$.

We present participants with both generations and a gallery with original art in the target style. We ask participants to decide which image is better in terms of style and quality, separately. For this, we ask them two different questions:

1. Based on noise, artifacts, detail, prompt fit, and your impression, which image has higher quality?
2. Overall, ignoring quality, which image better fits the style of the style samples?

For each comparison, we collect data from 5 users. We randomize several aspects of our study to minimize user bias. We randomly select the order of robust mimicry and baseline generations. Second, we randomly shuffle the order of all image comparisons to prevent all images from the same mimicry method to appear consecutively. Finally, we also randomly sample the seeds that models use to generate images to prevent repeating the same baseline image across different comparisons.

**Differences with Glaze's user study.** Our study does not exactly replicate the design of Glaze's user study for two reasons. First, the Glaze study provided annotators with four AI-generated images and four original images, asking if the generated images successfully mimicked the original artwork. This evaluation fails to account for the commonly encountered scenario where current models are incapable of reliably mimicking an artist's style even from unprotected art. Second, we believe the relative assessment recorded in our study ("Which of these two mimicry attempts is more successful?") is easier for humans than the absolute assessment used in the Glaze study ("Is this mimicry attempt successful").

**Prompts.** We curate a small dataset of 10 prompts $\mathbb{P}$. We design the prompts to satisfy two criteria:

1. *The prompts should cover diverse motifs with varying complexity.* This ensures that we can detect if a scenario compromised the prompt-following capabilities of a style mimicry model.
2. *The prompts should only include prompts for which our finetuning base model* M, *SD 2.1, can successfully generate a matching image.* This reduces the impact of potential human bias against common defects of SD 2.1.

To satisfy criterion 1 and increase variety, we instruct ChatGPT to generate prompt suggestions for four different categories:

1. *Simple prompts* with template "a {subject}".
2. *Two-entity prompts* with template "a {subject} {ditransitive verb} a {object}".
3. *Entity-attribute prompts* with template "a {adjective} {subject}".
4. *Entity-scene prompts* with template "a {subject} in a {scene}".

The chat we used to generate our prompts can be accessed at https://chatgpt.com/share/ea3d1290-f137-4131-baca-2fa1c92b3859. To satisfy criterion 2, we generate images with SD 2.1 on prompts suggested by ChatGPT and manually filter out prompts with defect generations (e.g. a horse with 6 legs). We populate the final set of prompts $\mathbb{P}$ with 4 simple prompts, 2 two-entity prompts, 2 entity-attribute prompts, and 2 entity-scene prompts (see Figure 29).

```
1  prompts = [
2      # simple prompts
3      "a mountain",
4      "a piano",
5      "a shoe",
6      "a candle",
7      # two-entity prompts
8      "a astronaut riding a horse",
9      "a shoe with a plant growing inside",
10     # entity-attribute prompts
11     "a feathered car",
12     "a golden apple",
13     # entity-scene prompts
14     "a castle in the jungle",
15     "a village in a thunderstorm",
16 ]
```

Figure 29: Our set of prompts. We manually wrote the prompts "a astronaut riding a horse" and "a village in a thunderstorm". ChatGPT wrote the remaining prompts.

**Quality control.** We first run a pilot study where we directly ask users to answer the previous questions about style and quality. This study resulted in very low-quality responses that are barely better than random choice. We enhanced the study to introduce several quality control measures to improve response quality and filter out low-quality annotations:

1. We limit our study to desktop users so that images are sufficiently large to perceive artifacts introduced by protections.

2. We precede the questions we use for our study with four dummy questions about the noise, artifacts, detail, and prompt matching of the images. The dummy questions force annotators to pay attention and gather information useful to answer the target questions.

3. We precede our study with a *training session* that shows for question 1, 2, and each of the four dummy questions an image pair with a clear, objective answer. The training session helps users to understand the study questions. We introduced this stage after gathering valuable feedback for annotators.

4. We add *control comparisons* to detect annotators who did not understand the tasks or were answering randomly. We generated several images from the baseline model trained on the original art. For each of these images, we created two ablations. For question 1 (quality), we include Gaussian noise to degrade its quality but preserve the same information. For question 2 (style), we apply Img2Img to remove the artist style and map the image back to photorealism using the prompt *"high quality photo, award winning"*. We randomly include control comparisons between the original generations and these ablations, and we only accept labels from users who answered correctly at least 80% of the control questions.

**Execution.** We execute our study on Amazon Mechanical Turk (MTurk). We design and evaluate an MTurk Human Intelligence Task (HIT) for each artist $A \in \mathbb{A}$, shown in Figure 30. Each HIT includes image pair comparisons for a single artist $A$ under all scenarios $\mathcal{S} \in \mathbb{M}$, as well 10 quality control image pairs, 10 style control image pairs, and 6 training image pairs. We generate an image pair for each of the 10 prompts and each of 15 scenarios, for a total of $10 \cdot 15 + 10 + 10 + 6 = 176$ image pairs per HIT. We estimate study participants to spend 5 minutes on the training image pairs and 30 seconds per remaining image pair, so 90 minutes in total. We compensate study participants at a rate of \$16/hour, so \$24 per HIT.

### K.1 STYLE MIMICRY SETUP VALIDATION

We execute an additional user study to validate that our style mimicry setup in Appendix G successfully mimics style from unprotected images.

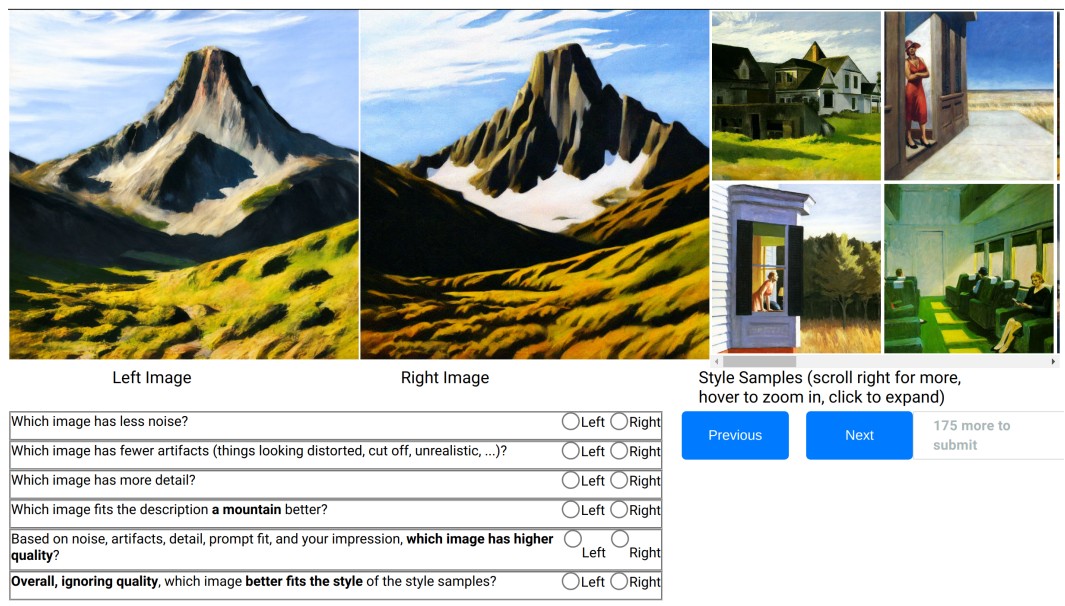

Figure 30: The interface of our user study.

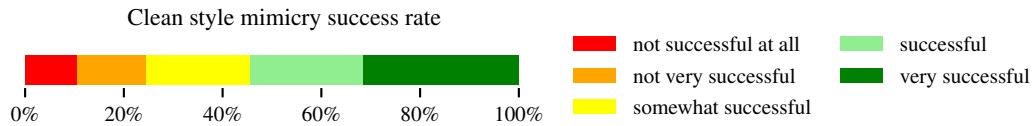

Figure 31: User ratings of clean style mimicry success. Each bar indicates the percentage of votes for the corresponding success level for clean style mimicry generations. Figure 32 breaks the ratings down by artist.

For each prompt $P \in \mathbb{P}$ and artist $A \in \mathbb{A}$, our validation study uses the baseline model trained on uprotected art to generate one image. Inspired by the evaluation by Glaze (Shan et al., 2023a), we ask participants to evaluate the style mimicry success by answering the question:

> How successfully does the style of the image mimic the style of the style samples? Ignore the content and only focus on the style.

To answer this question, we show a participant the image $x_A^O$ and the images $X^{\mathbb{A}}$ that serve as style samples. The participant can answer the question on a 5-point Likert scale with options

1. Not successful at all
2. Not very successful
3. Somewhat successful
4. Successful
5. Very successful

We also execute the style mimicry validation study on MTurk. We design and evaluate a single HIT for all questions, shown in Figure 33. We estimate study participants to spend 15 seconds on each question, and to spend 1 minute to familiarize themselves with a new style, so 35 minutes in total. We compensate study participants at a rate of $18/hour, so $10.50 per HIT.

We find that style mimicry is successful in over 70% of the comparisons. Results are detailed in Figure 31.

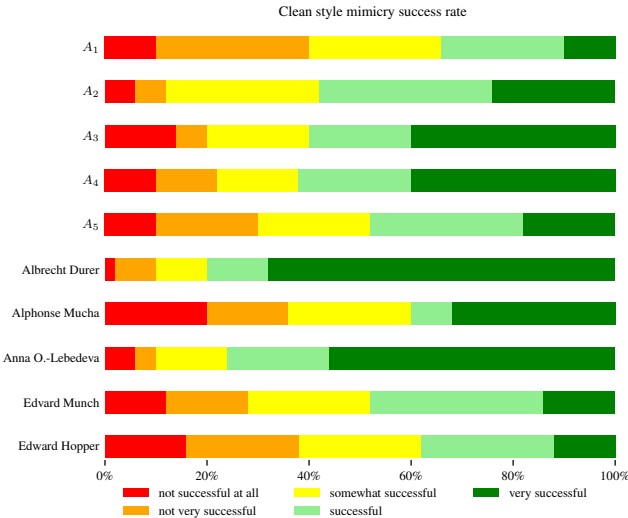

Figure 32: User ratings of clean style mimicry success. Each bar indicates the percentage of votes for the corresponding rate on the overall degrated mimicry generation for the corresponding artist.

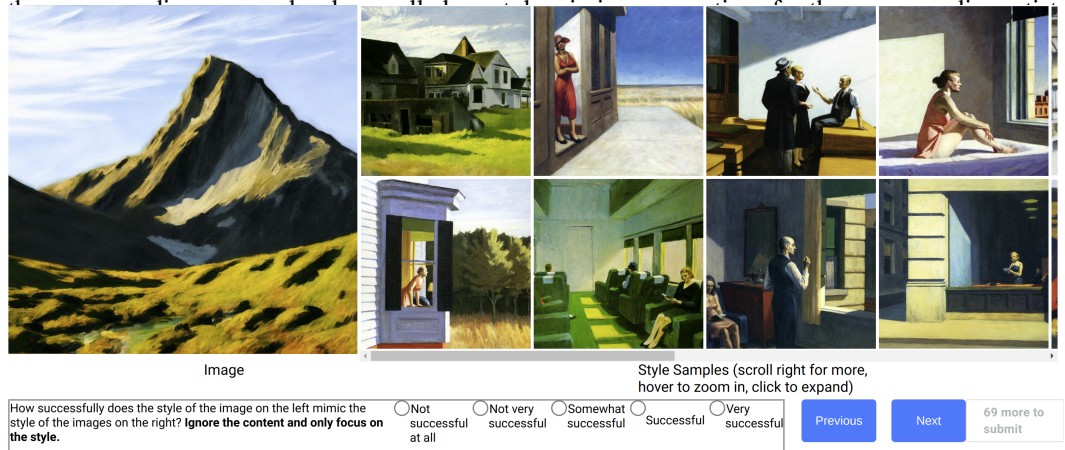

Figure 33: The interface of our style mimicry setup validation study.

## K.2 DOES PRE-PROCESSING ALONE DEGRADE IMAGE QUALITY?

While purification methods can nullify the effects of adversarial purifications, they could, in principle, also degrade image quality. To evaluate the extent of this phenomenon, we include comparisons between artists' original art, and their original art pre-processed with Noisy Upscaling. We include these comparisons for six artists[14] in our study and add comparisons for two held-out original artworks for each artist. On average, participants preferred the quality of pre-processed originals exactly 50 % of the time, and their style 48.3 % of the time. This suggests that Noisy Upscaling does not meaningfully degrade the quality of original artwork.

---

[14]We only include six out of the ten artists, because this experiment was added while the study was already ongoing.

## L    COMPUTE RESOURCES

Table 4 reports the compute resources for our experiments.

Table 4: Compute resources for our experiments. *Execution time per image / (artist)* reports the execution time of the method to compute a single image, or the combined execution time for all samples of an artist, if the method operates on all samples of an artist at once. † Google Cloud ‡ IMPRESS++ requires an additional 2 seconds per image generation.

| Method | GPU | CPU | Memory | Storage | Execution time per image / (artist) | Overall execution time |
|---|---|---|---|---|---|---|
| Finetuning | RTX A6000 | EPYC 7742 | 5 GB | 5 GB | (40 minutes) | 100 hours |
| Image generation | RTX A6000 | EPYC 7742 | 5 GB | 5 GB | 15 seconds | 13 hours |
| Anti-DB | RTX A6000 | EPYC 7742 | 5 GB | 10 GB | 29 minutes | 88 hours |
| Glaze | T4 | 16 vCPUs on GCP† | 5 GB | 5 GB | 4 minutes | 12 hours |
| Mist | RTX A6000 | EPYC 7742 | 5 GB | 5 GB | 18 seconds | 54 minutes |
| Gaussian noising | None | EPYC 7742 | 0 GB | 0 GB | 143 milliseconds | 26 seconds |
| IMPRESS++ | RTX A6000 | EPYC 7742 | 5 GB | 5 GB | (27 minutes)‡ | 370 minutes‡ |
| DiffPure | RTX A6000 | EPYC 7742 | 7 GB | 7 GB | 48 seconds | 144 minutes |
| Noisy Upscaling | RTX A6000 | EPYC 7742 | 3.5 GB | 3.5 GB | 217 seconds | 651 minutes |

