# OpenReview forum: "Adversarial Perturbations Cannot Reliably Protect Artists From Generative AI"
_ICLR.cc/2025/Conference — ICLR 2025 Spotlight_

### Official Review · Reviewer_ZKox · 2024-10-18

**Soundness:** 3
**Presentation:** 3
**Contribution:** 3
**Rating:** 8
**Confidence:** 4

**Summary:**

This paper studies the practicality of "using adversarial perturbations to protect art works from mimicry". The paper argues that, existing research works which uses adversarial noise for art copyright protection -- although published in top ML/Security conferences -- do not robustly achieve their claimed goals. Simple technical means are enough to bypass these existing protections. The work argues that researchers and practitioners should rethink the solution for art mimicry problem and develop alternative protections.

**Strengths:**

+ S1: The motivation is clear and compelling, effectively serving the paper's purpose: to caution researchers about the limitations of using adversarial perturbations for protection against art mimicry.

+ S2: The paper is well-organized and easy to follow. The experiments conducted are solid.

+ S3: The conclusions drawn from this work are potentially significant for the community and could reshape the landscape of this research field. Researchers are encouraged to reconsider the current paradigms of artwork / copyright data protection in light of their practical effectiveness.

**Weaknesses:**

+ Typos
    1. Line 219, fare -> fail?
    2. Figure 11, 13, 14, 15 seem to have the wrong label - now they are all labelled as "gaussian noising"

**Questions:**

Overall, the technical details in the paper are clearly described, so I have no questions about the technical aspects. However, I do have some open questions for the authors that I would like to discuss:

+ Q1: Practical and scalable evaluations? The core claim of this paper is to demonstrate the ineffectiveness of existing art mimicry protection methods. For this purpose, using a user study as an evaluation method is sufficient. However, for future papers proposing alternative protective solutions, what could be a scalable way to evaluate their effectiveness?

+ Q2: Is it technically possible to protect artists from mimicry? The conclusions drawn from this work appear to be somewhat pessimistic. Given that generative AI has a strong capability to fit any content, it seems that the potential for mimicry might be inevitable. What could be potential solutions to mitigate this issue?

---

> ### Author Response · Authors · 2024-11-20
>
> We thank the reviewer for their time and comments. We have fixed all the typos in the updated versions and answer the questions next:
>
> > Q1: Practical and scalable evaluations? The core claim of this paper is to demonstrate the ineffectiveness of existing art mimicry protection methods. For this purpose, using a user study as an evaluation method is sufficient. However, for future papers proposing alternative protective solutions, what could be a scalable way to evaluate their effectiveness?
>
> Ideally, future works on protections will adhere to adaptive evaluations best practices [1]. Thus, our methods (or novel purification strategies) can directly be included within the original evaluation and benefit from a standardized experimental design to facilitate replication and comparison. In the future, automated methods may become a more reliable substitute of human judgment, enabling more scalable evaluations.
>
> [1] Carlini, Nicholas, and David Wagner. "Adversarial examples are not easily detected: Bypassing ten detection methods." Proceedings of the 10th ACM workshop on artificial intelligence and security. 2017.
>
> > Q2: Is it technically possible to protect artists from mimicry? The conclusions drawn from this work appear to be somewhat pessimistic. Given that generative AI has a strong capability to fit any content, it seems that the potential for mimicry might be inevitable. What could be potential solutions to mitigate this issue?
>
> Indeed, we believe that technical solutions like adversarial perturbations will never reliably protect all artists. Non-technical solutions may prove viable, but we do not feel qualified to discuss non-technical solutions. ML researchers might be able to inform new laws or policies by clarifying what a technology can (or cannot) do, but these solutions are ultimately outside the realm of science.

---

> > ### Comment · Reviewer_ZKox · 2024-11-25
> > **Thank you for your rebuttal**
> >
> > Thank you and I will keep my score as accept.

---

### Official Review · Reviewer_pXVf · 2024-10-23

**Soundness:** 2
**Presentation:** 4
**Contribution:** 3
**Rating:** 6
**Confidence:** 4

**Summary:**

The paper presents a user study evaluating pre-processing techniques against current adversarial perturbations, finding that such methods can significantly reduce their effectiveness.

**Strengths:**

1. The writing is clear and easy to follow.
2. The topic is interesting and important.

**Weaknesses:**

The main weakness is the unreliable evaluation and whether the experimental results reflect real-world cases.

1. The study heavily relies on MTurk workers, who may not be suitable for this task. They might lack the expertise to classify artistic styles or assess whether artwork quality has been degraded. Artists themselves, such as those mentioned in [1], should be included in the main evaluation. Additionally, these workers may not represent potential consumers of artists' works, contrary to what the authors propose (L 338). A more thorough study should report the workers' relevance to the art domain (e.g., how often they purchase art or visit museums).
2. The effectiveness of the fine-tuning procedure is unclear. As shown in Fig. 3 (especially the bottom row) and Fig. 5, the generated images, even without protection, do not closely resemble the style of the training images. In contrast, previous work like Glaze [1] uses a fine-tuning setting with much stronger resemblance between generated images and training data (see Fig. 2 in [1]). The focus should be on whether adversarial perturbations effectively defend against mimicry once it has been successfully achieved. Even in Appendix K, where the limitations of using MTurk for evaluation are acknowledged, fewer than 50% of cases were rated as better or equal to “successful” mimicry.

    In an extreme case, if fine-tuning involved only one step, adversarial perturbations would likely fail to defend against mimicry. The core issue seems to be underestimating how closely the model need to fit training images, including adversarial perturbations,  to learn the style, which may result in an underestimation of their impact.

3. Even if the above weaknesses are overlooked, the paper provides no new insights on solving the problem of mimicry. Both pre-processing and adversarial perturbations degrade image quality, raising the question of whether removing adversarial perturbations is worth the cost, given that pre-processing might degrade the image quality in ways that hinder style recognition.
4. The paper overlooks broader ethical implications, such as the removal of adversarial perturbations used as a watermark. Similar to traditional watermarks (such as a simple icon on the corner) that can be removed but are illegal to do so in many countries, stronger watermarks complicate removal and leave evidence. A discussion on these ethical issues would contribute more meaningfully to the community.
5. The proposed methods lack novelty, largely offering improved hyper-parameter tuning of existing approaches.

[1] Shan S, Cryan J, Wenger E, et al. Glaze: Protecting artists from style mimicry by {Text-to-Image} models[C]//32nd USENIX Security Symposium (USENIX Security 23). 2023: 2187-2204.

**Questions:**

See weaknesses above.

---

> ### Author Response · Authors · 2024-11-20
>
> We thank the reviewer for their time and comments. We address the weaknesses as follow:
>
> > The study heavily relies on MTurk workers, who may not be suitable for this task. They might lack the expertise to classify artistic styles or assess whether artwork quality has been degraded. Artists themselves, such as those mentioned in [1], should be included in the main evaluation.
>
> We disagree with the fact that MTurk annotators are not suitable for this task. We have several baseline experiments that provide results similar to those obtained by the Glaze user study with artists (see Appendix K). Also, we train our annotators and filter them with strict quality questions that less than 20% of annotators passed. Appendix K includes all details on the design.
>
> > Additionally, these workers may not represent potential consumers of artists' works, contrary to what the authors propose (L 338). A more thorough study should report the workers' relevance to the art domain (e.g., how often they purchase art or visit museums).
>
> Studying consumer behavior is outside the scope of our work. As we detail above, we have carefully designed our human study to match baselines in the original Glaze work (where they used artists as judges), and believe they serve as a good evaluation for how humans perceive AI-generated art.
>
> > The effectiveness of the fine-tuning procedure is unclear. As shown in Fig. 3 (especially the bottom row) and Fig. 5, the generated images, even without protection, do not closely resemble the style of the training images. In contrast, previous work like Glaze [1] uses a fine-tuning setting with much stronger resemblance between generated images and training data (see Fig. 2 in [1]).
>
> We disagree with this assessment. In Appendix D, we compare our fine-tuning setting against the fine-tuning setting used in Glaze. Our results may even indicate improved performance with our fine-tuning setting, but in any case we believe the two are definitely comparable. The cross-paper figure comparison suggested by the reviewer features completely different styles, and is thus not suited to judge the fine-tuning settings.
>
> > The focus should be on whether adversarial perturbations effectively defend against mimicry once it has been successfully achieved. Even in Appendix K, where the limitations of using MTurk for evaluation are acknowledged, fewer than 50% of cases were rated as better or equal to “successful” mimicry.
>
> This is incorrect: Appendix K shows that the majority, i.e. more than 50% of cases were rated as “successful” or “very successful” mimicry. In addition, less than a quarter of cases were rated as “not very successful” or “not successful at all”.
>
> > Even if the above weaknesses are overlooked, the paper provides no new insights on solving the problem of mimicry. Both pre-processing and adversarial perturbations degrade image quality, raising the question of whether removing adversarial perturbations is worth the cost, given that pre-processing might degrade the image quality in ways that hinder style recognition.
>
> Our study provides the crucial insight that, in the eyes of humans, current mimicry protections can be (nearly) completely disabled.
>
> > The paper overlooks broader ethical implications, such as the removal of adversarial perturbations used as a watermark. Similar to traditional watermarks (such as a simple icon on the corner) that can be removed but are illegal to do so in many countries, stronger watermarks complicate removal and leave evidence. A discussion on these ethical issues would contribute more meaningfully to the community.
>
> As ML researchers, our work is to inform that protections are not effective in all cases. Our findings might inform new laws or policies by clarifying what a technology can (or cannot) do, but these discussions are ultimately outside the realm of science.
>
> > The proposed methods lack novelty, largely offering improved hyper-parameter tuning of existing approaches.
>
> We emphasize that most methods we implement already existed before the protections themselves were released. We demonstrate that original evaluations were incomplete and that previous methods could bypass protections. The novelty of our work is quantifying this effect for the first time via a human study. In addition, we are the first to evaluate the highly effective Noisy Upscaling as a technique for purification.

---

> ### Comment · Reviewer_pXVf · 2024-11-21
> **Thank you for detailed reply.**
>
> > similar to those obtained by the Glaze user study with artists (see Appendix K)
>
> I'm puzzling about what is the exact similarity here and further clarification could be good. As there is no comparison with Glaze user study in the given Sec.
>
> > Studying consumer behavior is outside the scope of our work.
>
> **It is the authors who mentioned** "consumer behavior" in their original paper to strengthen the how MTurk can be suitable for the task. "However, we believe that the judgment of non-artists is also relevant as they may ultimately represent potential consumers of digital art." I want to highlight nothing about consumer behavior but about **whether the augment is convincing or not** without any signal about connections between these workers with art.
>
> > In Appendix D, we compare our fine-tuning setting against the fine-tuning setting used in Glaze.
>
> Need any kind of quantative results to show it. There is only a result with one fig each fine-tuning method in Sec D.
>
> > Appendix K shows that the majority
>
> Appendix K **does not** show majority. Let's just count it. 5 out of 10 given classes in Fig 32, A1, A5, Alphonse Mucha, Edward Munch, Edward Hopper not getting with more than half of attack viewed as more or equal than successful. This should be tuned down to avoid misleading.
>
> No new reply on "pre-processing might degrade the image quality".
>
> > these discussions are ultimately outside the realm of science
>
> These are indeed connecting to the the paper as the paper is trying to discuss topic lies in the connection of techniques with society. Though this is pretty ok if there is no such discussion as this is not a necessity.

---

> ### Author Response · Authors · 2024-11-21
> **response**
>
> > I'm puzzling about what is the exact similarity here and further clarification could be good. As there is no comparison with Glaze user study in the given Sec.
>
> We cannot compare the studies' responses one-to-one since the Glaze study used a different set of responders, and only released aggregate statistics.
>
> But if we look at those aggregate statistics (Fig 31 in our paper vs Fig 9 in the Glaze paper) we find that artists and our respondents rate Glaze's mimicry as successful at similar rates. (note that Fig 9 in Glaze reports protection success rates while we report the opposite, successful mimicry rates).
>
> More precisely, our respondents said mimicry was "very successful" in ~30% of cases, "successful" in ~25% of cases, and "somewhat successful" in ~20% of cases, for a total of ~75% of cases where the mimicry is viewed as at least somewhat successful.
>
> In comparison, the Glaze study respondents say that without any specific protection, mimicry protection is "not (very) successful" (which we interpret as the mimicry succeeding at least somewhat) in ~75% of cases.
> So, on aggregate, our respondents and the Glaze respondents view mimicry attempts (with undefended images) as similarly successful.
>
> > Appendix K does not show majority. Let's just count it. 5 out of 10 given classes in Fig 32, A1, A5, Alphonse Mucha, Edward Munch, Edward Hopper not getting with more than half of attack viewed as more or equal than successful. This should be tuned down to avoid misleading.
>
> We were referring to Figure 31, which as stated above shows that our respondents view mimicry as at least somewhat successful in 75% of cases for undefended images, and successful or very successful in 55% of cases.
>
> If we look at it per artist, there are indeed some artists where mimicry is more successful than for others. This is consistent with what the Glaze paper says (we cannot compare results directly as the Glaze paper only gives aggregate results across all artists).
>
> > Need any kind of quantative results to show it. There is only a result with one fig each fine-tuning method in Sec D.
>
> We pointed to Appendix D in response to the reviewer's original qualitative claim that our finetuning results looked worse than those in Glaze.
>
> Our main quantitative argument here is sketched above: when we ask our user-study respondents to rate mimicry successes with our finetuning script, they rate mimicry as successful at a similar rate as the respondents in the Glaze user-study.
>
> > No new reply on "pre-processing might degrade the image quality".
>
> Sorry about missing this. We actually ran this comparison in our user-study. Specifically, we asked respondents to compare images before and after pre-processing to see if it has any effect on perceived image quality.
>
> Our results indicate that these two setups were indistinguishable to respondents, thus giving strong evidence that our pre-processing has negligible effect on image quality. We have added this important comparison to the paper (see Appendix K.2).
>
> > I want to highlight nothing about consumer behavior but about whether the augment is convincing or not without any signal about connections between these workers with art.
>
> Our argument is that MTurk respondents (after heavy filtering) are a reasonable cohort for our study, due to two reasons:
> 1. On aggregate, they view mimicry attempts as successful at a similar rate to the artists surveyed in Glaze (see above)
> 2. The perception of mimicry from non-artists also matters in our opinion.
>
> We have quantitative evidence in favor of (1). (2) is indeed an opinion, and something we don't see how to validate in any quantitative manner.

---

> ### Comment · Reviewer_pXVf · 2024-11-21
> **The comparison is fair.**
>
> I think with the details provided, the paper now includes results comparable to Glaze's setup. I hope all the information that matches the Glaze setup can be mentioned in the revised paper, as the paper is directly targeting the user study setup to align with Glaze, claiming Glaze-like defenses may not be as promising as suggested.
>
> Regarding the second main question about evaluation using MTurk workers, it is not as ideal as having artists conduct the evaluation, as artists are the ones who ultimately apply these adversarial watermarks. However, with the information highlighting results that align with Glaze (based on artists' measurements), I can assume this might serve as a rough estimation of artists' evaluation. Still, this difference needs to be emphasized, particularly in how MTurk workers' results align with artists' evaluation in glaze.
>
> Based on this information, I have increased my score to 6. I hope the points we discussed here can be partially included in the revised paper, as it is critically important to verify whether we are conducting real and fair comparisons and measurements, especially when claiming that a line of methods is not effective—particularly in a context where a user study is involved.

---

> > ### Author Response · Authors · 2024-11-21
> > **thanks**
> >
> > We thank the reviewer for the fruitful discussion.
> > We will make sure to revise the paper to highlight these points

---

### Official Review · Reviewer_SUG2 · 2024-10-26

**Soundness:** 3
**Presentation:** 4
**Contribution:** 3
**Rating:** 8
**Confidence:** 4

**Summary:**

This paper examines the effectiveness of existing style protection methods (such as Glaze, Mist, and Anti-DreamBooth) against both black-box and white-box attacks. The authors demonstrate that these protections can be easily circumvented using simple techniques like DiffPure, upscaling, and adding Gaussian noise, as well as more sophisticated methods like IMPRESS. The findings are supported by an MTurk study, where participants were asked to distinguish between images generated using style mimicry on unprotected vs protected artworks. The paper highlights the inherent asymmetry between attackers (style mimickers) and defenders (artists), cautioning against relying on adversarial perturbations for style protection due to the potential false sense of security they may provide.

**Strengths:**

1.  While it was anticipated that prominent style protection techniques like Glaze might have weaknesses, the extent of their fragility is striking. The paper demonstrates that these methods fail even against rudimentary attacks, with Glaze unable to withstand a mere change in the fine-tuning script. This revelation underscores a concerning lack of “security mindset” within the style protection research community, which is particularly alarming given the recognition Glaze has received, including multiple awards from USENIX Security.

2. The MTurk study is well constructed and the authors have taken pains to ensure that their work adheres to commonly accepted ethical standards.

**Weaknesses:**

1. The work has limited novelty since it was predictable that style protection would be vulnerable to DiffPure-like purification methods. It would be good if the purification methods evaluated in the paper could be packaged into a standard baseline, say on the lines of AutoAttack.

2. In my opinion, the paper overstates the general case against style protection techniques based on adversarial perturbation. The presented argument makes an assumption that artists have the choice to release their artworks on the internet, and they may choose to withhold their artworks if they believe that AI models may be trained on their artworks. However, digital artists are extremely dependent on the internet to grow their customer pool and advertise their works, so this is likely not a feasible option. The appropriate counterfactual to artists using Glaze-like methods would be not using any protections at all. Also, since these methods seem to be improving against simple attacks at least, it may be enough to use them as a deterrent rather than as fool-rpoof security.


I struggled to decide whether to rate this paper as borderline accept or clear accept due to the stated weaknesses. Ultimately, I decided to rate this paper as clear accept as the argument may indeed be persuasive to a small minority of artists who may decide to not publish any of their works if there is no secure style protection mechanism.

**Questions:**

1. It would be interesting to see the results on Glaze 2.1 (the newest version) for completeness’s sake. Does this induce greater quality degradation compared to its previous versions?

2. The “Best-of-4” method is not really a practical method since it depends on the human ratings and the same human ratings are used to evaluate the method. For fair comparison, the raters need to be split into validation and test raters, and the method selection should utilize only the validation raters and be evaluated by the test raters, with averaging across splits via cross-validation.

---

> ### Author Response · Authors · 2024-11-20
>
> We thank the reviewer for their time and comments, and have updated the paper to incorporate their feedback. We address the weaknesses and clarifications as follow:
>
> > The work has limited novelty since it was predictable that style protection would be vulnerable to DiffPure-like purification methods.
>
> We fully agree that the highlighted vulnerabilities _should_ have been predicted by the proposed defenses. However, we are the first work doing a human study to measure the effects of different pre-processing techniques.
>
> > It would be good if the purification methods evaluated in the paper could be packaged into a standard baseline, say on the lines of AutoAttack
>
> We recognize the value of a standard AutoAttack-like baseline, but do not currently see a way to automatically tune and select the best purification methods, because existing automated style mimicry metrics are not sufficiently reliable, and human labels are required. We will make our code open-source and hope that future protections adversarially evaluate their methods.
>
> > In my opinion, the paper overstates the general case against style protection techniques based on adversarial perturbation. The presented argument makes an assumption that artists have the choice to release their artworks on the internet, and they may choose to withhold their artworks if they believe that AI models may be trained on their artworks. However, digital artists are extremely dependent on the internet to grow their customer pool and advertise their works, so this is likely not a feasible option. The appropriate counterfactual to artists using Glaze-like methods would be not using any protections at all. Also, since these methods seem to be improving against simple attacks at least, it may be enough to use them as a deterrent rather than as fool-rpoof security.
>
> We agree with this. However, hiding vulnerabilities and providing artists with a false sense of security is not the way to go. We believe there is value in informing artists about the risks associated for them to make informed decisions.
>
> > It would be interesting to see the results on Glaze 2.1 (the newest version) for completeness’s sake. Does this induce greater quality degradation compared to its previous versions?
>
> Glaze 2.0 was released as a response to our research, see Figures 20 and 21 for qualitative examples. We have obtained similar results for Glaze 2.1. In absence of a full human evaluation, we believe that the qualitative examples for 2.0 serve as a good illustration.
>
> > The “Best-of-4” method is not really a practical method since it depends on the human ratings and the same human ratings are used to evaluate the method. For fair comparison, the raters need to be split into validation and test raters, and the method selection should utilize only the validation raters and be evaluated by the test raters, with averaging across splits via cross-validation.
>
> With the “Best-of-4” method, we intend to simulate the scenario where a style mimicrist has finetuned several models, one with each robust mimicry method. The mimicrist can now prompt each model, and choose the image that they like the most. It also illustrates that different methods succeed for different styles and artists. We have made an update to the paper to reflect this.

---

> > ### Comment · Reviewer_SUG2 · 2024-11-20
> > **Acknowledged**
> >
> > Thanks for the rebuttal, I will keep my score unchanged.

---

### Official Review · Reviewer_JzLh · 2024-11-01

**Soundness:** 4
**Presentation:** 3
**Contribution:** 3
**Rating:** 8
**Confidence:** 4

**Summary:**

The authors critically revisit the current efforts towards protecting artists' work from diffusion model mimicry. The author proposes that most protection nowadays cannot really protect artists, since the protection can by easily bypassed using some tricks of purifications e.g. Gaussian Noising, Diff-Pure and Up-scalers. Extensive experiments are done to support the claim in this paper. Overall, the paper is well-written, easy to follow and the proposed method is simple but effective.

**Strengths:**

- The paper is well-written and easy to follow.
- It works on an important problem and provides critical insights: all protection methods today cannot protect artworks from diffusion-based mimicry. Though works like Glaze have been widely accepted by artists, they actually perform bad.
- The proposed robust mimicry methods are simple but effective.
- The authors did extensive experiments to support the claims.

**Weaknesses:**

Clarification:
- The title of this paper is 'ADVERSARIAL PERTURBATIONS CANNOT RELIABLY PROTECT ARTISTS FROM GENERATIVE AI', while style-based mimicry is one part of diffusion-based mimicry, there are more basic mimicry e.g. inpainting, style-transfer by diffusion model, which are also tested in previous papers of protection e.g. Mist. Fine-tuning a diffusion model seems to have a more complicated mechanism compared with image-to-image applications of diffusion models.
- I wonder does the proposed method also work for inpainting/image-to-image SDEdit, if it works, the proposed method becomes more general.

Methods:
- I noticed that the perturbation used in this paper is quite small, if the noise is scaled up, will the purification be worse
- While Glaze and Mist are popular, there are many other protection methods that can be studied to get a safer conclusion. e.g. MetaCloak [3] and SDS [4].

Related Papers:
[1, 2] are highly related to this paper, [1] also find that the current attacks are vulnerable to purifications, [2] proposed that latent diffusion models can be easily purified by pixel-space diffusion models.

[1] Can Protective Perturbation Safeguard Personal Data from Being Exploited by Stable Diffusion?

[2] Pixel is a Barrier: Diffusion Models Are More Adversarially Robust Than We Think.

[3] MetaCloak: Preventing Unauthorized Subject-driven Text-to-image Diffusion-based Synthesis via Meta-learning

[4] Toward effective protection against diffusion-based mimicry through score distillation

**Questions:**

See weaknesses.

---

> ### Author Response · Authors · 2024-11-20
>
> We thank the reviewer for their time and comments. We clarify the questions as follow:
>
> > Does the proposed method also work for inpainting/image-to-image SDEdit, if it works, the proposed method becomes more general.
>
> We limit our evaluation to finetuning, since it has been identified as the strongest mimicry method [1]. However, we believe our purification will exhibit similar properties for all methods as they operate at pre-processing and remove the effects of the protections to begin with.
>
> > I noticed that the perturbation used in this paper is quite small, if the noise is scaled up, will the purification be worse
>
> We actually use strong protections in our evaluation. We explain the choice of our perturbation budget in Appendix J.2. Quoting the paper:
> > For a fair comparison, we fix the perturbation budget p for each adversarial perturbation δx created by Mist and Anti-DreamBooth to p = 8/255, which is the same budget that Liang et al. use to evaluate Mist. [...] Glaze is closed-source software whose perturbation budget control only offers the settings Default, Medium, and High. Upon request, the Glaze authors refused to share a codebase where we could control the hyperparameters. Therefore, we evaluate Glaze through their official public tool with the setting High to evaluate our protections under the highest protections.
>
> [1] Shan S, Cryan J, Wenger E, et al. Glaze: Protecting artists from style mimicry by {Text-to-Image} models[C]//32nd USENIX Security Symposium (USENIX Security 23). 2023: 2187-2204.

---

> > ### Comment · Reviewer_JzLh · 2024-11-26
> > **thanks for the reply**
> >
> > Thanks for addressing most of my concerns. I have some following questions here:
> >
> > - I notice that you also use DiffPure to purify the noise. I am confused since the noise (e.g. Mist) is designed to fool the SDEdit, will the DiffPure just generate bad outputs instead doing the purification?

---

> > > ### Author Response · Authors · 2024-11-26
> > > **Response**
> > >
> > > While the Mist paper claims protection against DiffPure, their evaluation is based on an outdated DiffPure implementation
> > > that applies a diffusion model trained on ImageNet. In contrast, we apply DiffPure with Stable Diffusion XL, a significantly stronger diffusion model. According to our evaluation, DiffPure applied with Stable Diffusion XL is relatively successful in purifying artwork (see Figure 4 for the user study, Figure 8b for a purification sample, and Figure 13 for mimicry samples).
> > >
> > > For reference, see also Section 4.1 in our paper:
> > > > For instance, Mist (Liang et al., 2023) evaluates against DiffPure purifications
> > > using an outdated and low-resolution purification model. Using DiffPure with a more recent model,
> > > we observe significant improvements.

---

> > > > ### Comment · Reviewer_JzLh · 2024-11-27
> > > > **response**
> > > >
> > > > If the noise is generated to attack SDXL, will the DiffPure fail?

---

> ### Author Response · Authors · 2024-11-27
> **response**
>
> We only evaluate adversarial perturbations against SD 2.1, as this is the model that all considered mimicry protections evaluate against. However, it is quite likely that adversarial noise targeting specifically SDXL would reduce the effectivity of DiffPure applied with SDXL. We deliberately use a different diffusion model, since attackers can freely choose from different existing models to bypass protections, with new models released continually, and as discussed in the paragraph "Protections are broken from day one, and cannot improve over time": once protected art is published online it cannot be updated retroactively since old versions will likely remain accessible by adversaries.

---

### Meta-Review · Area_Chair_a1wj · 2024-12-20

**Metareview:**

This paper revisits the solution of "using adversarial perturbations to protect artworks from mimicry". The paper argues that existing research works that use adversarial noise for art copyright protection do not robustly achieve their claimed goals. Simple technical means are enough to bypass these existing protections. Hence, the work indicates that researchers and practitioners should rethink the solution for the art mimicry problem and develop alternative protections. Reviewers agree that this work has the following advantages: (1) It works on an important problem and provides critical insights. It demonstrates that even the well-known protection work has serious weaknesses (2) The proposed robust mimicry methods are simple but effective. (3) The work did extensive experiments to support the claims. (4) The writing is clear and easy to follow. The reviewers also raise some questions: (1) Extension to other generation models like inpainting/image-to-image SDEdit. (2) Extension to other protection methods. (3) Limited novelty. All concerns are well responded by authors during the rebuttal process.

**Additional Comments On Reviewer Discussion:**

All reviewers provide meaningful and solid comments for the works. Three reviewers rate the paper a clear accept and one reviewer gives the borderline accept. The main concerns are (1) Extension to other generation models like inpainting/image-to-image SDEdit. (2) Extension to other protection methods. (3) Limited novelty. All concerns are well responded by authors during the rebuttal process.

---

### Decision · Program_Chairs · 2025-01-22

Accept (Spotlight)